# Combined targeting of G protein-coupled receptor and EGF receptor signaling overcomes resistance to PI3K pathway inhibitors in PTEN-null triple negative breast cancer

Davide Zecchin[1], Christopher Moore[1], Fanourios Michailidis[1], Stuart Horswell[2], Sareena Rana[1,3], Michael Howell[4] & Julian Downward[1,3,*] iD

## Abstract

Triple-negative breast cancer (TNBC) has poorer prognosis compared to other types of breast cancers due to the lack of effective therapies and markers for patient stratification. Loss of PTEN tumor suppressor gene expression is a frequent event in TNBC, resulting in over-activation of the PI 3-kinase (PI3K) pathway and sensitivity to its inhibition. However, PI3K pathway inhibitors show limited efficacy as monotherapies on these tumors. We report a whole-genome screen to identify targets whose inhibition enhanced the effects of different PI3K pathway inhibitors on PTEN-null TNBC. This identified a signaling network that relies on both the G protein-coupled receptor for thrombin (PAR1/F2R) and downstream G protein βγ subunits and also epidermal growth factor receptor (EGFR) for the activation of the PI3K isoform p110β and AKT. Compensation mechanisms involving these two branches of the pathway could bypass PI3K blockade, but combination targeting of both EGFR and PI3Kβ suppressed ribosomal protein S6 phosphorylation and exerted anti-tumor activity both *in vitro* and *in vivo*, suggesting a new potential therapeutic strategy for PTEN-null TNBC.

**Keywords** G protein; p110β; PTEN; resistance; triple-negative breast cancer
**Subject Categories** Cancer; Pharmacology & Drug Discovery; Signal Transduction

## Introduction

Triple-negative breast cancer (TNBC) is defined by the lack of expression of the actionable markers estrogen receptor (ER), progesterone receptor (PR), and human epidermal growth factor receptor 2 (HER2). It accounts for about 15% of all breast cancer. There are no targeted therapies currently available in the clinic for the treatment of TNBC besides chemotherapy (Chacon & Costanzo, 2010; Bianchini *et al*, 2016). Despite being chemosensitive, TNBC is characterized by a short time to progression and by the poorest prognosis among the other subtypes of breast cancers (Liedtke *et al*, 2008). The aggressiveness and heterogeneity of this disease, together with the current paucity of therapeutic options, suggests a need for new molecular markers for patient stratification and new targeting approaches for the treatment of TNBC.

PTEN deficiency occurs in up to 35% of TNBC (Cancer Genome Atlas Network, 2012), representing one of the most commonly altered tumor suppressor genes in this subtype of cancer. PTEN is a lipid phosphatase that dephosphorylates PtdIns$(3,4,5)P_3$ to form PtdIns$(4,5)P_2$ and counteracts the enzymatic activity of PI3K (Maehama & Dixon, 1998). Loss of PTEN results in accumulation of PtdIns$(3,4,5)P_3$ at the inner surface of the plasma membrane and over-activation of AKT (Stambolic *et al*, 1998; Haddadi *et al*, 2018). Consistently with its effect at the signaling level, PTEN-deficient TNBCs show up-regulation of markers of activation of PI3K pathway, such as phospho-AKT (both Thr308 and Ser473), phospho-mTOR, phospho-p70$^{S6K}$, phospho-S6, and phospho-4EBP1 (Stemke-Hale *et al*, 2008; Cancer Genome Atlas Network, 2012). Pre-clinical evidence also showed that PTEN-null tumors, including breast cancers, are sensitive to the inhibition of PI3K pathway and in particular to the targeting of specific nodes of the signaling route, such as the β isoform of PI3K (Jia *et al*, 2008; Wee *et al*, 2008; Hancox *et al*, 2015) and AKT (Chen *et al*, 2006; Vasudevan *et al*, 2009; Sangai *et al*, 2012).

Drugs targeting multiple components of the PI3K pathway, including PI3Kβ and AKT inhibitors, are in clinical trials (reviewed in (Janku *et al*, 2018)) and may represent a rational therapeutic

---

1 Oncogene Biology, Francis Crick Institute, London, UK
2 Computational Biology, Francis Crick Institute, London, UK
3 Lung Cancer Group, Institute of Cancer Research, London, UK
4 High Throughput Screening Laboratories, Francis Crick Institute, London, UK
*Corresponding author. Tel: +44 20 3796 1838; E-mail: julian.downward@crick.ac.uk

opportunity to treat the subgroup of TNBCs characterized by PTEN deficiency (Delaloge & DeForceville, 2017). The success obtained employing PI3K isoform-specific inhibitors in the treatment of relapsed chronic lymphocytic leukemia (CLL) (Furman *et al*, 2014), relapsed indolent lymphoma (Gopal *et al*, 2014), and PIK3CA-mutant breast cancers (Juric *et al*, 2018a,b) demonstrated the clinical potential of inhibiting specific nodes of PI3K signaling in patients selected based on cancer type and biomarkers. However, apart from those specific contexts, PI3K pathway inhibitors generally delivered only modest effects in the clinical setting (Janku *et al*, 2018), and no clear evidence of benefits for PTEN-deficient tumor patients has been reported so far (Kim *et al*, 2017; Martin *et al*, 2017). While pre-clinical data indicate the requirement for PI3K pathway activity for the survival and proliferation of PTEN-null TNBC cells, clinical evidence suggests the need to inhibit additional targets to enhance the effect of PI3K pathway inhibitors, widen their therapeutic window, reduce their toxicity, and produce a sustained anti-tumor effect.

We set out to interrogate the whole genome to identify additional targets whose inhibition might synergize with inhibitors of the PI3K pathway. We focused on drugs affecting different nodes of the PI3K pathway, with particular attention to PI3Kβ isoform, pan-PI3K and AKT inhibitors. Our investigation revealed a previously unappreciated complexity of compensation mechanisms that impairs the response to PI3K pathway inhibitors in PTEN-deficient TNBC. This knowledge has paved the way to the rational design of possible combinatorial targeting strategies for the effective treatment of this type of cancer.

# Results

### A whole-genome shRNA screen identified EGFR inhibition as enhancer of the response to PI3K pathway inhibitors in triple-negative breast cancer cells

A number of pre-clinical models showed that PTEN-deficient cancer cells are sensitive to inhibition of PI3K pathway, and this was particularly clear when p110β or AKT were selectively targeted (Chen *et al*, 2006; Jia *et al*, 2008; Wee *et al*, 2008; Vasudevan *et al*, 2009). However, in most of the cases, PI3K pathway inhibitors only partially impaired the proliferation of PTEN-null cancer cells (Fig EV1A). In order to identify genes whose silencing could enhance the effect of PI3K and AKT inhibitors, we performed a whole-genome short hairpin (sh) RNA interference screen in combination with drugs targeting the PI3K pathway, including an isoform-specific p110β/δ inhibitor (AZD8186), a pan-PI3K class I inhibitor (GDC0941), and an AKT inhibitor (MK2206). AZD8186 inhibits with high affinity both p110β and p110δ isoforms (IC50 = 4 nM for p110β and 12 nM for p110δ). However, PTEN-null triple-negative human breast cancer (TNBC) cells, the main focus of our study, do not express appreciable levels of p110δ, as shown by comparing expression levels of the genes encoding p110 isoforms in these cells and in B-cell acute lymphoblastic leukemia lines (Fig EV1B, https://portals.broadinstitute.org/ccle), that express all p110 isoforms (Thorpe *et al*, 2015). It is therefore likely that the effects observed following treatment by AZD8186 in PTEN-null TNBC cell lines are due to targeting of p110β, rather than p110δ.

---

**Figure 1. A short-hairpin screening identified hits whose inhibition enhanced the effect of PI3K pathway inhibitors on PTEN-deficient triple-negative breast cancers.**

A   Schematic representation of the shRNA screening aimed to identify genes whose knock-down enhanced the anti-proliferative effects of AZD8186 (PI3Kβi), GDC0941 (pan-PI3Ki), or MK2206 (AKTi) on the PTEN-null cell line MDA-MB-468. The shRNAs directed against the transcripts of three genes are shown in three different colors. shRNAs against gene 1 are depleted in the vehicle-treated condition compared to time 0, and they most likely target an essential gene. shRNAs against the transcripts of gene 2 are unchanged in all conditions, while shRNAs silencing gene 3 are selectively depleted in the AZD8186-selected cells compared to vehicle and their abundance is not affected by the 2-week long culture with vehicle. Gene 3 is a good candidate for inhibition in combination with AZD8186.

B   Dot-plot showing the fold change (log$_2$) in number of reads between vehicle- and AZD8186-treated conditions vs the *P*-value of the difference between the two treatment conditions for each shRNA. 9 out of 18 shRNAs targeting EGFR showed a *P*-value < 0.2 and are highlighted in the plot. The plot was generated considering the results from biological triplicate of the experiment.

C, D   MDA-MB-468 was infected with the indicated shRNAs targeting EGFR and selected by puromycin. EGFR mRNA was then measured by RT–qPCR (C), and cell viability was measured after 4 days of treatment with serial dilutions of AZD8186 (D). Average $\pm$ SD of triplicates and representative of three independent experiments.

E, F   MDA-MB-468 was treated with serial dilutions of gefitinib (E) or lapatinib (F) in the presence of vehicle, AZD8186 (0.25 μM), GDC0941 (0.25 μM), or MK2206 (0.45 μM), as indicated. Cell viability was measured after 4 days and normalized within each of the PI3K pathway inhibitor-treated condition to the viability in the absence of gefitinib or lapatinib. Average $\pm$ SD of triplicates and representative of two independent experiments.

G   Viability of six PTEN-null vs five PTEN-WT TNBC cell lines—not carrying other known mutations in PIK3CA, PIK3CB, or PIK3R1 genes—treated with PI3Kβi (AZD8186 90 nM), EGFRi (gefitinib 3 μM) alone, or in combination for 6 days. Mean of 3 independent experiments $\pm$ SD. Statistical significance of two-tailed unpaired student *t*-test in PTEN-null PI3Kbi vs PI3Kbi + EGFRi **$P$ = 0.0059, PTEN-null EGFRi vs PI3Kbi + EGFRi **$P$ = 0.0047, PTEN-null PI3Kbi + EGFRi vs PTEN-WT PI3Kbi + EGFRi *$P$ = 0.0459, PTEN-WT PI3Kbi vs PI3Kbi + EGFRi *$P$ = 0.0108, PTEN-WT EGFRi vs PI3Kbi + EGFRi n.s. $P$ = 0.1926. PTEN-null cell lines used in the experiments were as follows: MDA-MB-468, HCC70, HCC1937, HCC38, HCC1395, and BT-549; PTEN-WT cell lines were as follows: MDA-MB-157, MDA-MB-231, HCC1187, HCC1428, and HCC1806.

H   Synergy score for combinations of serial dilutions of PI3Kβi (AZD8186) plus EGFRi (gefitinib) tested on six PTEN-null and five PTEN-WT TNBC cell lines for 6 days in three independent experiments. The score was obtained analyzing the viability data through the software Chalice Analyser. Mean of the synergy scores $\pm$ SD. Statistical significance of Mann–Whitney two-tailed test *$P$ = 0.0303.

I   Patient samples from METABRIC dataset classified as TNBCs were assigned to the groups "PTEN-low" when falling in the lower quartile for PTEN expression, "PTEN-mut" when harboring a non-synonymous mutation on PTEN gene, "PTEN-WT" in all other cases. Comparison of the expression of EGFR between the PTEN low or mut and the PTEN-WT groups. Data presented in a box and whisker plot with the central band indicating the median, the upper, and lower extremes of the box or hinge being the third and first quartiles, respectively, and the whiskers extending to the most extreme data values. Mean $\pm$ SD. $P$-value calculated by unpaired *t* test.

Source data are available online for this figure.

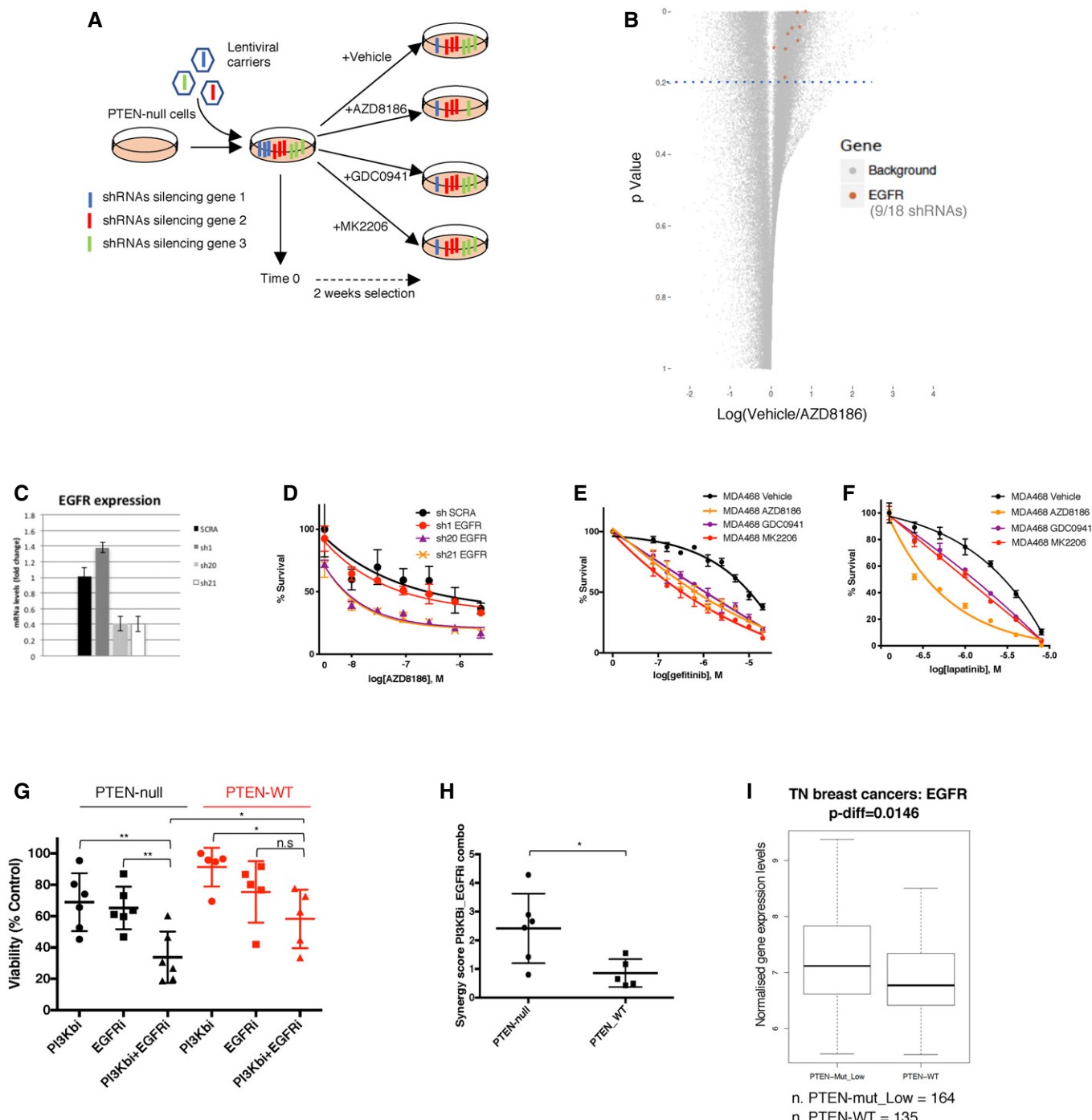

**Figure 1.**

As a cellular model, we employed the PTEN-deficient TNBC cell line MDA-MB-468. Also, EGFR is genetically amplified and overexpressed in this cell line, recapitulating molecular features of a significant fraction of TNBCs (Reis-Filho *et al*, 2006; Reis-Filho & Tutt, 2008; Gumuskaya *et al*, 2010; Shao *et al*, 2011; Martin *et al*, 2012; Park *et al*, 2014; Nakai *et al*, 2016). MDA-MB-468 exhibited average sensitivity to PI3K pathway inhibitors compared to other cell lines with similar genetic alterations and origin (Fig EV1A).

The cancer cells were infected with a library of lentiviral vectors expressing shRNAs targeting most of the genes encoded by the human genome (around 16,000 genes) in combination with either vehicle, AZD8186, GDC0941, or MK2206 treatments (Fig 1A). AZD8186 was employed at a concentration that selectively targeted p110β over p110α (Schwartz *et al*, 2015). Similar to AZD8186, also GDC0941, and MK2206 were used at concentrations close to the IC$_{30}$ for MDA-MB-468 cells (Fig EV1C), in order to allow the

silencing of specific genes to show synergistic effects in combination with the drugs.

After 14 days of drug selection, the genomic DNA of cells in different treatment conditions was extracted and sequenced to evaluate the relative abundance of the different shRNA species within the cellular populations. We initially filtered genes for which at least two shRNAs were selectively depleted in the treated compared to untreated condition (see Materials and Methods) and these genes were then ranked based on the effect (lowest treatment/control reads ratio) of their second-best performing shRNA (Table EV1). We identified nine shRNAs targeting EGFR that were consistently depleted in the AZD8186-treated condition compared to the vehicle control (Fig 1B), and the EGFR gene ranked within the top 1% candidates in all treatment conditions (Table EV1). We validated our findings by infecting MDA-MB-468 with three shRNAs targeting EGFR and previously included in the shRNA screening. Only two shRNAs were able to effectively silence the expression of EGFR (Fig 1C), and those two shRNAs were the ones showing a combination effect with AZD8186 treatment in impairing the proliferation of MDA-MB-468 (Fig 1D).

Next, we validated the cooperation between PI3K pathway and EGFR inhibition by the use of EGFR-targeted drugs. We observed increased activity in MDA-MB-468 for different EGFR inhibitors, including small molecules gefitinib, lapatinib, erlotinib, and the monoclonal antibody cetuximab, in the presence of AZD8186, but also in the presence of other inhibitors of PI3K pathway such as GDC0941 or MK2206 compared to vehicle control (Figs 1E and F and EV1D). It is noticeable that both genetic and pharmacological suppression of EGFR in MDA-MB-468 exerted only marginal anti-proliferative effect on its own. This is in line with the lack of response to EGFR inhibitors observed in clinical trials in triple-negative breast cancers (von Minckwitz *et al*, 2005; Dickler *et al*, 2008; Carey *et al*, 2012; Yardley *et al*, 2016). We also extended our findings using a panel of TNBC cell lines, comparing the effect of drug combinations on PTEN-deficient and PTEN-WT cell lines. All cell lines employed in these experiments did not harbor any known mutation in PIK3CA, PIK3CB, PIK3R1, or KRAS that may confound or modify the response to those drugs (COSMIC database). The combination of gefitinib and AZD8186, MK2206, or GDC0941 cooperatively reduced the cell viability in the panel of PTEN-null cells, while their effect on PTEN-WT models was considerably more limited (Figs 1G and, EV1E and F). Consistently, a higher synergistic score for the drug combination including AZD8186 and gefitinib was calculated in PTEN-deficient cell lines compared to WT (Fig 1H), showing that this combinatorial regimen is especially effective in a PTEN-null genetic background for triple-negative breast cancers. We also tested the combination of BYL719 (a p110α isoform-selective inhibitor) and gefitinib on three PTEN-null and three PTEN WT TNBC cell lines. In contrast to the combination with AZD8186, BYL719 combined with gefitinib did not produce any genotype-selective anti-proliferative effect on the PTEN-deficient cell lines, confirming that AZD8186 exerted its function by inhibiting p110β rather than p110α (Fig EV1G).

In order to test the relevance of our findings in patient samples, we asked whether EGFR was differentially expressed depending on PTEN status in the triple-negative breast cancer samples of the METABRIC database. We found that samples showing low expression and/or non-synonymous mutation of PTEN had statistically significant higher expression of EGFR compared to samples expressing higher level of WT PTEN (Fig 1I).

Through the analysis of the shRNA screening data, we also identified multiple shRNAs targeting the casein kinase 2 (CK2) components CSNK2B and CSNK2A2 as being depleted in the AZD8186-treated condition compared to vehicle (Fig EV1H). Those genes encode the regulatory (β) and one of the catalytic (α′) subunits of CK2 holoenzyme, respectively. The knock-down of CSNK2B by three of the shRNAs included in the screening showed a correlation between the degree of transcript down-regulation (Fig EV1I) and the enhancement of AZD8186 activity (Fig EV1J) in MDA-MB-468 cells, validating the results of the screening. We further confirmed those data by the use of CX-4945, a CK2 targeted kinase inhibitor (Fig EV1K). We observed that CX-4945 cooperatively reduced cell viability when combined with AZD8186, GDC0941, or MK2206 in a panel of PTEN-deficient TNBC cell lines, while the same effect was not found in PTEN-WT TNBC cells (Fig EV1L–N). Altogether, these data showed that EGFR or CK2 inhibition can potentiate the activity of PI3K pathway inhibitors selectively on those TNBCs lacking functional PTEN.

## Combinatorial targeting of PI3Kβ isoform and EGFR exerts anti-tumor effects on PTEN-null TNBC *in vivo*

We reasoned that simultaneous and specific inhibition of PI3Kβ isoform and EGFR may lead to a sustained synergistic anti-tumor effect also *in vivo* with a well-tolerated toxicity profile. This approach might be more tolerable than targeting both p110α and p110β using pan-PI3K inhibitors or inhibiting the downstream master regulator AKT. We evaluated the *in vivo* efficacy and the toxicity of the combination of AZD8186 and erlotinib on mice injected orthotopically in the mammary fat pads with the human cancer cells MDA-MB-468 or HCC70. These two cell lines both express high levels of EGFR, and they show different degree of sensitivity to AZD8186, GDC0941, and MK2206 *in vitro* (Fig EV1A). We observed in all cases no effect or only partial tumor growth inhibition for the single drug treatments. This was the case also for mice transplanted with HCC70, although those cells had previously shown higher sensitivity *in vitro* to AZD8186-mediated inhibition. The combination prevented tumor growth in MDA-MB-468 xenografts (Figs 2A and EV2A) and induced regression in HCC70 tumors (Fig 2B and C). The body weight of treated mice did not significantly change during single or combined treatments (Fig 2D), and no other signs of toxicity were detected, suggesting that the drug combination can be well tolerated *in vivo*.

Next, we asked whether the combined inhibition of PI3Kβ and EGFR may prove effective also in an immunocompetent context. We took advantage of a *Wap-cre:Pten*$^{fl/fl}$*:Tp53*$^{fl/fl}$ mouse model in which the expression of *Cre* by the *Wap* promoter drives the conditional inactivation of *Pten* and *Tp53* floxed alleles in the alveolar epithelial cells of the mammary glands of late pregnant and lactating female mice (Wagner *et al*, 1997). This, in turn, has been reported to induce the formation of triple-negative-like breast tumors with hyper-activated AKT signal and with an average latency of 9.8 months (Liu *et al*, 2014). We recreated the same genetic make-up in pure C57BL6/J background, in order to isolate tumors and derive cell lines that could be re-transplanted in the fourth mammary gland fat pad of immune-competent, syngeneic recipients. This strategy

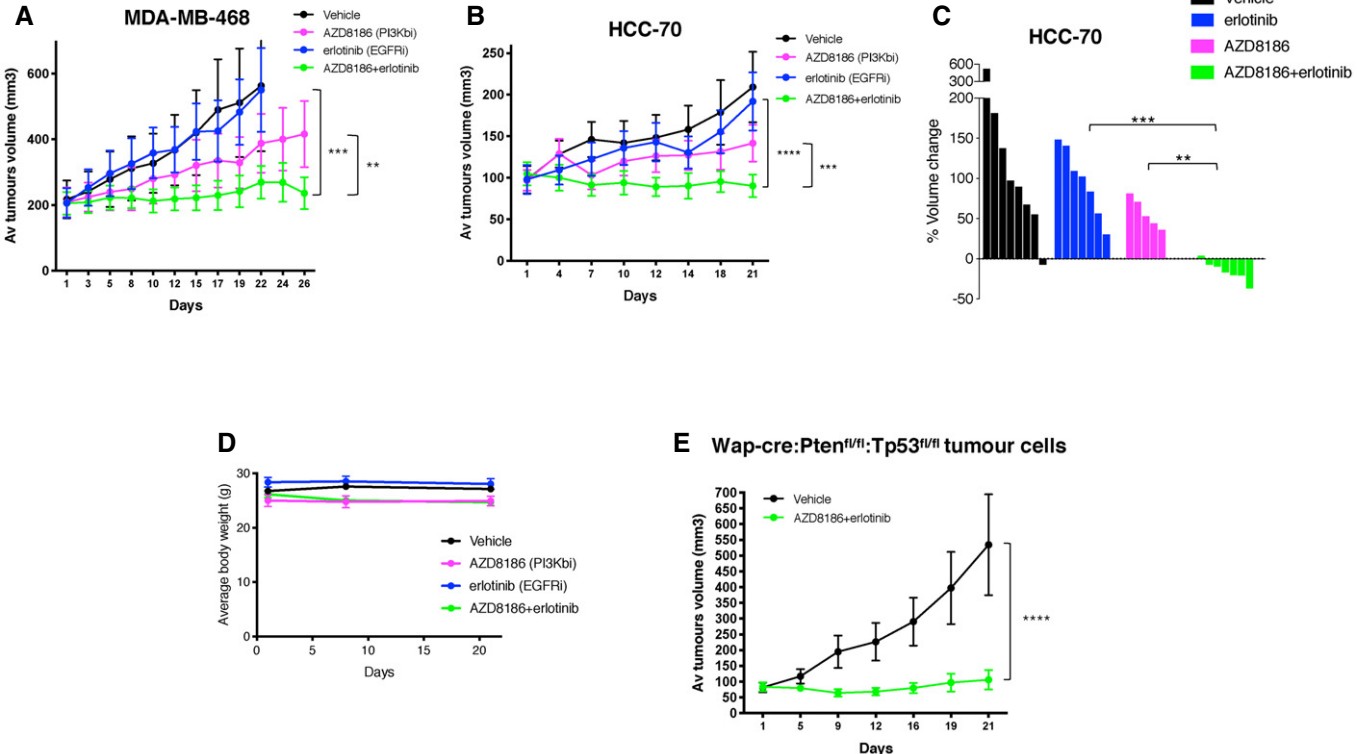

**Figure 2. Anti-tumor effects of combinatorial inhibition of PI3Kβ and EGFR on triple-negative breast cancers *in vivo*.**

A   Tumor volume of MDA-MB-468 mammary fat-pad xenografts treated with vehicle, AZD8186 (50 mg/kg, og twice/day), erlotinib (50 mg/kg IP once/day) alone, or in combination (5–6 mice per group, mean ± SEM). Statistical significance of two-way ANOVA statistical test **$P$ = 0.0028 and ***$P$ < 0.0001.

B   Tumor volume of HCC-70 mammary fat-pad xenografts treated with vehicle, AZD8186 (150 mg/kg, og once/day), erlotinib (50 mg/kg IP once/day) alone, or in combination (6–7 mice per group, mean ± SEM). Growth curves were compared using two-way ANOVA statistical test. Statistical significance of two-way ANOVA statistical test ***$P$ = 0.0006 and ****$P$ < 0.0001.

C   Waterfall representation of changes in the volume of individual HCC-70 tumors during the treatment. Statistical comparison between different treatment groups by Mann–Whitney test **$P$ = 0.0023 and ***$P$ = 0.0006.

D   Change in the body weight of mice harboring HCC70 xenograft tumors (6–7 mice per group, mean ± SEM) during 21 days of treatment.

E   A cell line derived from a mammary tumor spontaneously developed in a Wap-cre:Pten$^{fl/fl}$:Tp53$^{fl/fl}$ mouse was cloned and injected in the mammary fat pad of syngeneic C57BL6/J recipient mice. Tumors grew in 12 out of 35 transplanted mice, and only these tumors were selected for the treatments described in the figure. When tumors reached an average volume of 100 mm³, they were treated with vehicle or a combination of AZD8186 (150 mg/kg, og once/day) and erlotinib (50 mg/kg IP once/day). Tumors were then measured during the treatment (six mice per group, mean ± SEM). Statistical significance of two-way ANOVA statistical test ****$P$ < 0.0001.

Source data are available online for this figure.

allowed the generation of large cohorts of immune-competent mice harboring well-localized and synchronized transplanted tumors that could be challenged with different drug treatments.

We derived cell lines from a PTEN- and TP53-negative, EGFR-positive primary mammary tumor that developed in a *Wap-cre:Pten$^{fl/fl}$:Tp53$^{fl/fl}$* mouse and that was histologically classified as a carcinosarcoma resembling a spindle-cell, triple-negative type of tumor that can be found in the human breast (Fig EV2B). These cells showed a combinatorial response *in vitro* to treatment with AZD8186 and gefitinib (Fig EV2C), validating previous data obtained in human cancer cell lines. One of those clones was transplanted in the mammary fat pad of C57BL6/J female mice, and we observed engraftment of the injected cells in more than 95% of the cases. All mice were treated with vehicle, AZD8186, erlotinib, or a combination of the two drugs soon after engraftment of the cells (Fig EV2D). However, 2/3 of transplanted mice underwent

spontaneous tumor regression in the vehicle group. Single drug treatments were not effective in preventing the escape of a fraction of the treated tumors, while all tumors treated with combined AZD8186 and erlotinib showed clear regression.

We then selected out from a cohort of transplanted mice those tumors that were able to escape spontaneous regression, and we observed that the combined treatment with AZD8186 and erlotinib completely prevented the further aggressive growth of those isografts (Fig 2E). These results show that the combined inhibition of PI3Kβ and EGFR exerts anti-tumor effect on aggressive PTEN and TP53-null triple-negative-like breast tumor growth also in immune-competent models. Confirmation of anti-tumor activity and lack of toxicity for the drug combination in both immune-suppressed and immune-competent recipients also makes unlikely that AZD8186 (p110β/δ inhibitor) exerts its effects by targeting p110δ in the immune cell compartment.

### Decreased S6 phosphorylation is a marker of response to combinatorial treatments in PTEN-null triple-negative breast cancers

To define the mechanisms responsible for the cooperative impact on viability of the combinatorial regimens, we investigated which changes were induced by single and combined drug treatments at the biochemical level. Since PTEN deficiency is known to increase PI3K pathway activity and the drug combinations included an inhibitor of PI3K pathway, we initially focused on changes affecting this signaling route. We found that, as expected, the treatment with PI3Kβ inhibitor AZD8186 partially decreased the

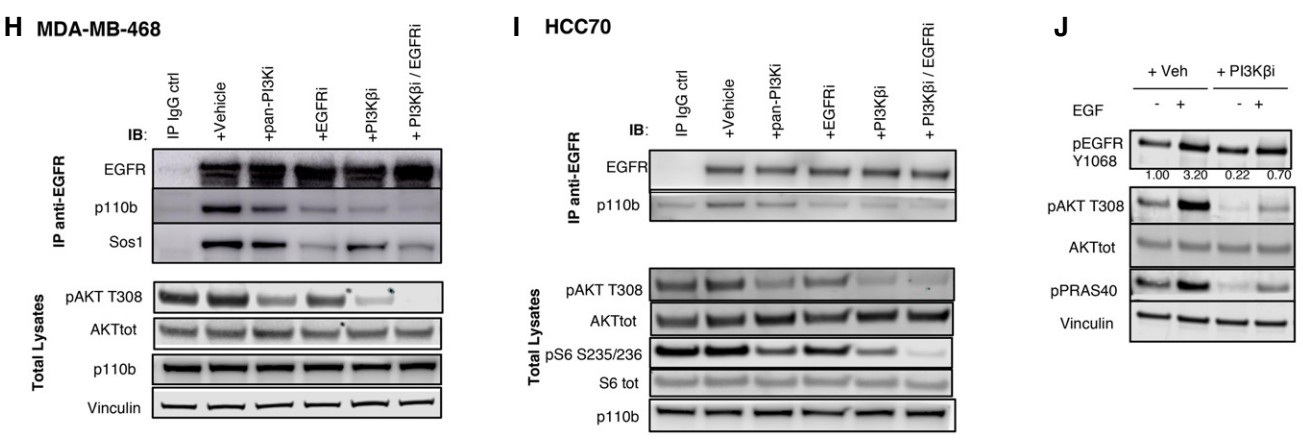

Figure 3.

**Figure 3. Biochemical effects of EGFR and PI3K pathway combined inhibition.**

A   MDA-MB-468 cells were treated for 24 h with vehicle, PI3Kβi (AZD8186 250 nM), EGFRi (gefitinib 3 μM), or cetuximab (100 μg/ml), alone or in combination. The cell lysates were probed with the indicated antibodies.

B   MDA-MB-468 cells were treated for 24 h with vehicle, AKTi (MK2206 450 nM), gefitinib (3 μM), or cetuximab (100 μg/ml), alone or in the indicated combinations. The cell lysates were probed with the indicated antibodies.

C, D   HCC70 (C) and ZR-75-1 (D) parental cells or PI3Kβi-Res (derivative cells with acquired resistance to AZD8186), or AKTi-Res (acquired resistance models to MK2206) were treated with vehicle, PI3Kβi (AZD8186 250 nM), or AKTi (MK2206 1 μM) for 24 h. The whole cell lysates were then probed with the indicated antibodies. Spliced images of parental and resistant paired samples were taken from the same original blot, and blots showing P-S6 and S6 tot in (D) have been cut and reassembled for figure purposes.

E   HCC70 parental, PI3Kβi-Res, or AKTi-Res cells were treated with vehicle, AKTi (MK2206 250 nM), or gefitinib (3 μM), alone or in the indicated combinations. The cell lysates were probed with the indicated antibodies.

F   HCC70 MK res (acquired resistance models to MK2206) were treated with serial dilutions of gefitinib, alone, or in combination with MK2206 (810 nM), as indicated, and viability measured after 4 days of treatment. Average ± SD of triplicates and representative of two independent experiments.

G   HCC70 AZD res (acquired resistance models to AZD8186) were treated with serial dilutions of gefitinib, alone, or in combination with AZD8186 (270 nM), as indicated, and viability measured after 4 days. Average ± SD of triplicates and representative of two independent experiments.

H, I   p110β co-immunoprecipitates with EGFR in MDA-MB-468 (H) and in HCC70 (I). Cells were pre-treated with different inhibitors, including vehicle, pan-PI3Ki GDC0941 (1 μM for MDA-MB-468 or 0.5 μM for HCC70), EGFRi (gefitinib 3 μM), PI3Kβi (AZD8186 250 nM), or a combination of EGFRi and PI3Kβi. Cell lysates were incubated with IgG control or anti-EGFR antibody, and the immuno-complexes or the total lysates were immune-blotted with the indicated antibodies.

J   EGF-induced increase in phospho-AKT is dependent on p110β kinase activity. MDA-MB-468 were starved in 0% FBS and pre-treated with vehicle or PI3Kβi (AZD8186 250 nM) for 1 h. Cell lysates were probed with the indicated antibodies. Phospho-AKT and pan-AKT bands were quantified by the use of ImageLite software: The ratio of phospho-AKT to pan-AKT normalized to the control (left hand lane) is shown.

Source data are available online for this figure.

phosphorylation of the downstream proteins AKT, PRAS40, and S6. Inhibition of EGFR reduced the levels of phospho-ERK independently of AZD8186, but only when combined with the PI3Kβ inhibitor it resulted in a more effective suppression of all the markers of PI3K pathway activation (Fig 3A). The same combination activity was observed when EGFR inhibitors were combined with the AKT inhibitor MK2206, especially at the level of the downstream marker phospho-S6 (Fig 3B). Similar biochemical effects following drug treatments were observed also in *Wap-cre: Pten^{fl/fl}:Tp53^{fl/fl}* mouse cell lines (Fig EV3A).

In order to determine whether the same changes described in cells only partially sensitive to inhibition of PI3K pathway were taking place also in models of acquired resistance, we derived drug-resistant variants from breast cancer cell lines that normally exhibit high sensitivity to PI3K inhibition *in vitro*, such as HCC70 and ZR-751. All acquired-resistant models were more resistant to PI3K pathway inhibition compared to the parental cells (Fig EV3B) and showed a less pronounced reduction in phospho-S6 following treatment with those drugs (Fig 3C and D). When EGFR inhibitors were combined with PI3K pathway inhibitors in acquired-resistant cells, we detected a stronger suppression of phospho-S6 (Fig 3E) and of cell viability (Fig 3F and G). These results suggest that some of the mechanisms that impaired the response to PI3K pathway inhibitors could be common between cell models with very different sensitivities to these drugs, including models of acquired resistance. However, in all cases the inhibition of S6 phosphorylation represented a good marker of response to treatments targeting PI3K pathway.

We also asked whether a synergistic suppression of phospho-S6 may underlie the impact on cell viability described for PI3K pathway and CK2 combined inhibition (Fig EV1I–N). We found that the simultaneous targeting of PI3Kβ and CK2, similarly to inhibition of PI3K pathway and EGFR, led to a stronger suppression of phospho-S6 (Fig EV3C). Altogether, these results enforced the notion that inhibition of phospho-S6 is a marker of response to combinatorial therapies including PI3K pathway inhibitors in PTEN-deficient triple-negative breast cancer cells.

**p110β signals downstream of EGFR in PTEN-deficient triple-negative breast cancer cells**

We next investigated which effectors mediated signaling downstream of EGFR in PTEN-null cells and how the interaction with those molecules was affected by drug treatments. We found that p110β, a key activator of the AKT pathway in PTEN-deficient tumor cells, co-immunoprecipitated with endogenous EGFR in two different TNBC cell lines (Figs EV3D and 3H–I). Sos1 and p110α were used as positive controls of the co-immunoprecipitation experiment, being known interactors of EGFR. The co-immunoprecipitation of EGFR and p110β was disrupted to different extents by pre-treatment with EGFR or PI3K inhibitors, while the combination between PI3Kβ and EGFR inhibitors was the most effective in impairing the interaction. This effect was mirrored by a stronger inhibition of AKT phosphorylation downstream, confirming our previous observations (Fig 3H and I). We asked whether the p110β-EGFR interaction was functional in the signaling triggered by EGFR. We observed that AZD8186 pre-treatment was able to impair both the basal and the EGF-induced phosphorylation of AKT (Fig 3J). These data demonstrate that EGFR can interact with p110β and that the activation of AKT induced by EGFR stimulation is, at least in part, dependent on p110β kinase activity in these cells. Also, effective drug-mediated inhibition of this interaction correlated with a stronger reduction in phospho-AKT and of downstream pathway activation.

We also investigated whether the synergistic reduction in PI3K pathway activity by combined PI3Kβ and EGFR inhibition was due to EGFR inhibitor's ability to suppress phospho-ERK. However, when we combined the treatment of PI3K pathway inhibitors with MEK inhibition, we did not detect any synergistic reduction in phospho-S6 (Fig EV3E), showing that the profound inhibition of phospho-S6 was more likely due to a stronger suppression of the upstream AKT activity.

**Targeting G protein β and γ subunits sensitizes to EGFR and pan-PI3K inhibitors**

With the aim of identifying and validating other modifiers of the response to PI3K pathway inhibitors, we carried out a CRISPR-Cas9-

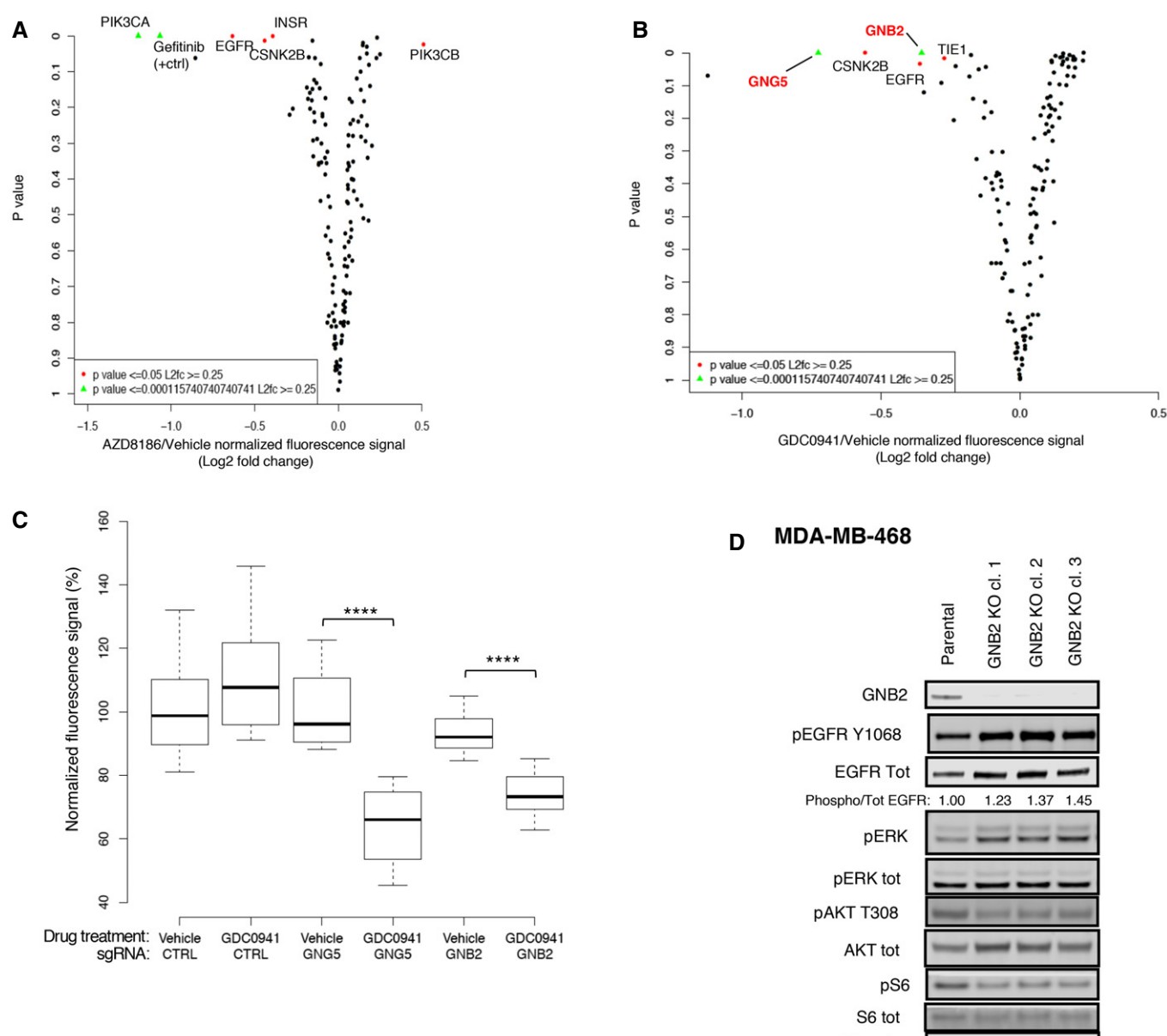

**Figure 4. A CRISPR-Cas9 screening identified GNB2 as a target to potentiate the inhibition mediated by pan-PI3K inhibitor.**

A, B   Results of CRISPR-Cas9 screening in combination with AZD8186 100 nM (A) or GDC0941 400 nM (B). The dot-plots show for each gene knocked-out by sgRNAs the fold change ($\log_2$) between treated conditions (AZD8186 or GDC0941, respectively) and vehicle in the fluorescence signal (anti-phosphoS6 immunofluorescence) vs the $P$-value of the difference calculated by two-sided $t$-test. The plots represent means of biological triplicates. Genes for which it was calculated a $P < 0.0001$ and $\log_2$ (fold change) > 0.25 are reported and highlighted in green; genes having a $0.0001 < P < 0.05$ and a $\log_2$ (fold change) > 0.25 are reported and shown in red. Gefitinib combined with AZD8186 represents the positive control of the experiment (A).

C   Box and whisker plot showing the fold change in fluorescence signal between vehicle and GDC0941-treated conditions for MDA-MB-468 cells transduced with non-target control, GNB2, or GNG5 sgRNAs ($N$ = 2 or 3). Data presented in a box and whisker plot with the central band indicating the median, the upper, and lower extremes of the box or hinge being the third and first quartiles, respectively, and the whiskers extending to the most extreme data values within 1.5 times the inter-quartile range. Statistical significance of unpaired $t$-test ****$P < 0.0001$.

D   Biochemical analysis of GNB2 KO cells. Cell lysates of MDA-MB-468 parental cells and three MDA-MB-468 GNB2 KO clones were probed with the indicated antibodies. Quantification of the bands was performed by ImageLite software.

Source data are available online for this figure.

mediated targeted screen of the top candidates from the genome wide shRNA screen, testing whether the knockout of these genes sensitized MDA-MB-468 to the inhibitory activity of AZD8186, GDC0941, or MK2206. We also added to the list of candidate genes those encoding proteins that were found to be regulated by AZD8186 treatment in MDA-MB-468 using a reverse-phase protein

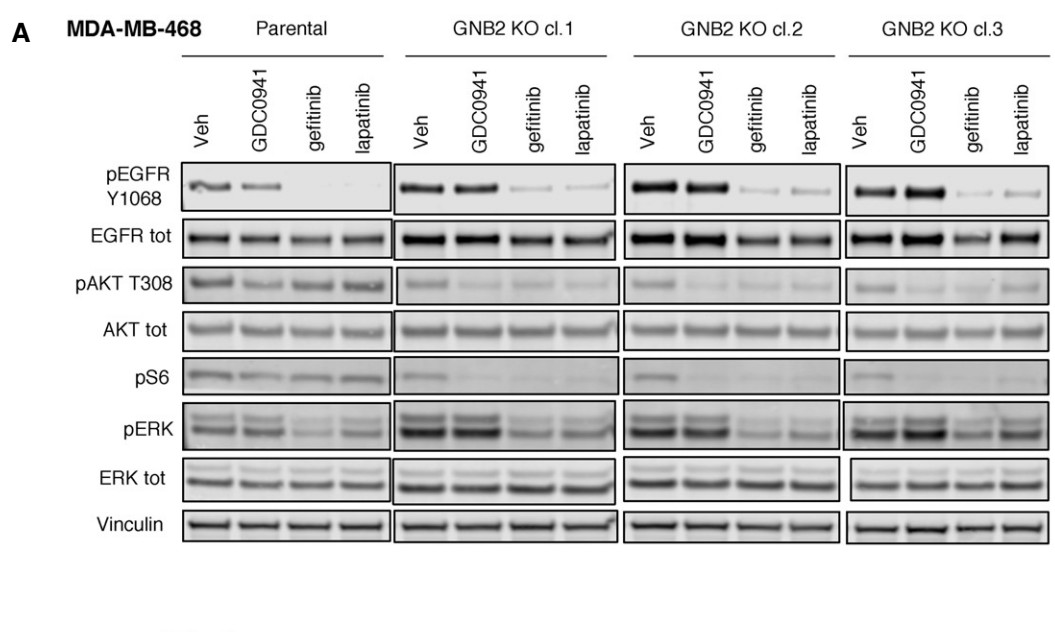

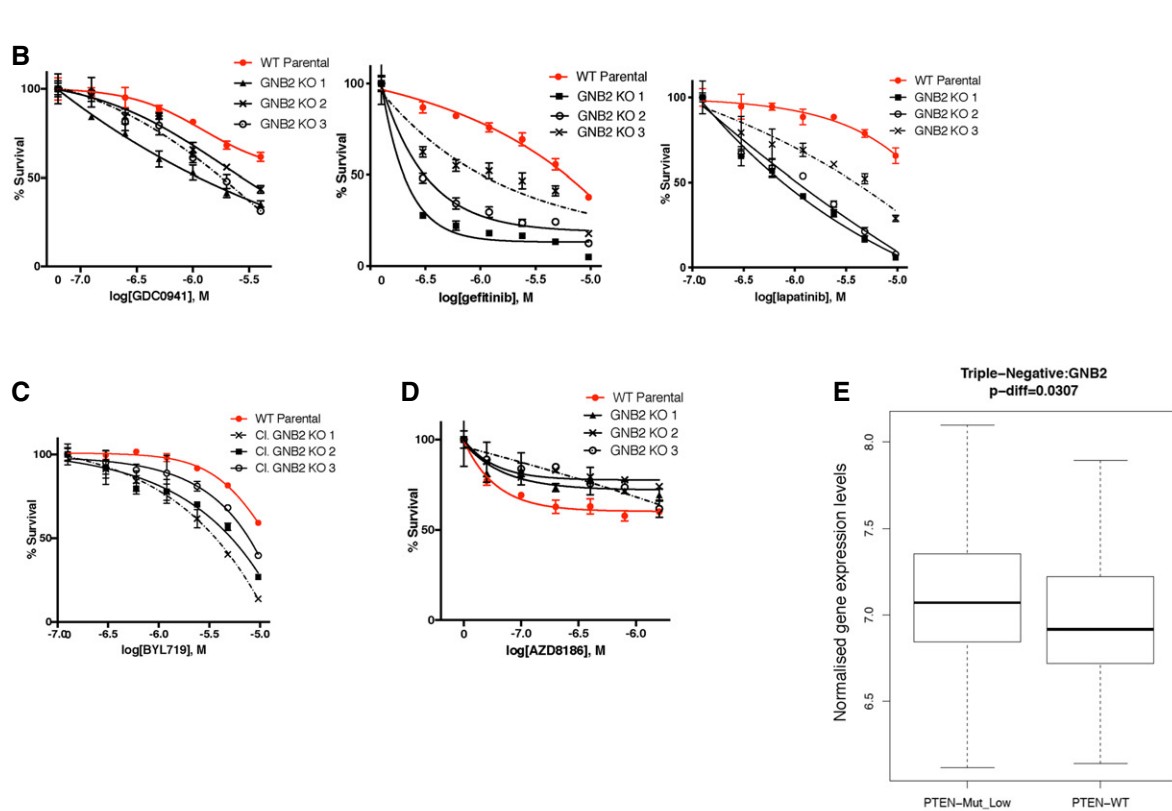

**Figure 5. GNB2 KO modifies the sensitivity to different inhibitors of EGFR-PI3K pathway.**

A   MDA-MB-468 parental cells and three MDA-MB-468 GNB2 KO clones were treated with vehicle, GDC0941 1 μM, gefitinib 3 μM, or lapatinib 1 μM for 24 h. The cell lysates were probed with the indicated antibodies.

B–D   Viability assays of MDA-MB-468 parental cells and three MDA-MB-468 GNB2 KO clones treated with serial dilutions of the indicated drugs for 6 days. Mean ± SD of triplicates and representative of two or three independent experiments.

E   Patient samples from METABRIC dataset classified as TNBCs (N = 299) were assigned to the groups "PTEN-low" when falling in the lower quartile for PTEN expression, "PTEN-mut" when harboring a non-synonymous mutation on PTEN gene, "PTEN-WT" in all other cases. Comparison of the expression of GNB2 between the PTEN low or mut and the PTEN-WT groups. Box and whisker plot (Median/IQR/1.5*IQR Whiskers). P-value calculated by unpaired t-test.

Source data are available online for this figure.

array (RPPA, Fig EV4A). We hypothesized, indeed, that some of these proteins may have a functional role in defining the response to PI3K pathway inhibitors. The list included overall 141 genes, of which 31 were from the RPPA data and 110 from the shRNA screen (Materials and Methods and Table EV2).

In order to perform the CRISPR-Cas9 screen, we derived and characterized a single-cell clone from an MDA-MB-468 population harboring a doxycycline-inducible expression vector for Cas9 (Fig EV4B–D). Given the importance of phospho-S6 inhibition as a marker of response to combinatorial drug regimens, we used the measurement of phospho-S6 by immuno-fluorescence as a readout of the screen. The screening was performed in an arrayed format, and each gene was efficiently targeted by a combination of 4–5 different sgRNAs (Fig EV4E–H). Plotting of the raw phospho-S6 signal from individual experiments showed overall consistent values among replicates in different treatment conditions (Fig EV4I), confirming the quality of the screening data.

The CRISPR-Cas9 screen in combination with AZD8186 confirmed our previous results, showing enhanced suppression of phospho-S6 by the combined treatment with AZD8186 and the KO of EGFR or CSNK2B (Fig 4A). Also, the KO of PIK3CA (encoding PI3K p110α) potentiated the effect of the PI3Kβ inhibitor. This confirmed data from previous literature that showed how simultaneous inhibition of both p110α and p110β exerted synergistic effect on PTEN-null cancer cells (Schwartz et al, 2015). On the other hand, knockout of PIK3CB, encoding PI3K p110β, the target of AZD8186, abolished the effect of the drug on phospho-S6 and provided a further validation to our approach.

Interestingly, two genes encoding for G protein β and γ subunits, GNB2 and GNG5, respectively, ranked among the top cooperative partners of GDC0941 when they were knocked-out (Fig 4B and C). In order to investigate the mechanistic basis of this interplay, we isolated GNB2 KO clones by immunofluorescence-based detection of GNB2 protein levels (Fig EV4K) and we compared signaling in the KO cells with parental MDA-MB-468 cells (Fig 4D). We found that the KO clones had an increased phosphorylation and expression of EGFR compared to the WT counterpart and they showed also an increased ratio between phosphorylated and total EGFR, suggesting a compensatory increase in EGFR activity. This finding was supported by the increased basal phosphorylation of the downstream effector ERK1/2. Also, we observed that markers of PI3K pathway activity such as phospho-AKT or phospho-S6 were decreased by the KO of GNB2, showing that this protein had an impact on the basal activation of this pathway. We then treated WT and GNB2 KO cells with GDC0941, and we observed a stronger and prolonged suppression of phospho-AKT and phospho-S6 in the KO clones compared to WT cells (Figs EV5A and 5A), validating the results of the CRISPR-Cas9 screening. Downregulation of the basal PI3K signaling and increased sensitivity to the GDC0941-mediated suppression of phospho-S6 was also observed in an independent TNBC cell line following the KO of GNB2 (Fig EV5B).

Interestingly, the increased activity of EGFR in cells lacking GNB2 was mirrored by a stronger suppression of PI3K signaling following EGFR inhibition by gefitinib or lapatinib (Fig 5A). As expected, also in viability assays GNB2 KO clones showed increased sensitivity to both GDC0941 and EGFR or HER family inhibitors, demonstrating a shift in the addiction to EGFR-PI3K axis following the disruption of G protein signaling (Fig 5B).

We asked then which PI3K isoform signals downstream EGFR to guarantee the survival of the cells in the absence of GNB2. We found that the KO of GNB2 increased the dependence of the cells on p110α compared to the WT counterpart, as demonstrated by treatment of WT and GNB2 KO cells with the p110α-specific inhibitor BYL719 (Fig 5C). GNB2 loss, on the other hand, partially relieved the dependence of PTEN-null cells on p110β (Fig 5D), and this shift was supported at the biochemical level by the differential impact of PI3K isoform-specific inhibitors on the phosphorylation of AKT and S6 downstream EGFR in WT or GNB2-deficient cells (Fig EV5C).

As both PI3K isoforms were partially involved in supporting the survival and the activation of PI3K pathway downstream EGFR in the absence of GNB2, these results explained the increased sensitivity of GNB2 KO cells to pan-PI3K inhibitors. The basal down-regulation of PI3K pathway activity and the shift in dependence following GNB2 inactivation from p110β to EGFR and to other downstream non-p110β effectors, such as p110α, are in line with the accepted notion that βγ subunits of G proteins are responsible for the activation of PI3Kβ (Kurosu et al, 1997) and suggested that this arm of the pathway can operate in parallel with EGFR to support the activation of AKT.

Altogether, these data suggested that both the βγ subunits of G proteins and EGFR can signal to AKT through PI3Kβ and that those two arms of the pathway can compensate each other when PI3K inhibitors are employed to treat PTEN-deficient TNBC cells.

We investigated the relevance of these findings in patients by analyzing the expression of GNB2 in TNBC samples divided according to their PTEN status in the METABRIC dataset. Consistent with the importance of G protein signaling in the PTEN-null context, we found statistically significant higher expression of GNB2 in samples characterized by low expression or non-synonymous mutation of PTEN compared to the PTEN-WT subgroup (Fig 5E).

## PAR1 signals to AKT through the βγ subunits of G proteins

Since we identified G protein βγ subunits as important activators of the PI3K pathway in PTEN-null cells, we asked which G protein-coupled receptors (GPCRs) can activate G protein signaling in these cells. Assuming that the inhibition of any GPCR acting upstream of the βγ subunits of G proteins in these cells should reproduce the phenotype described after knocking out the downstream effector GNB2, we tested a library of compounds targeting GPCRs for their ability to suppress phospho-S6 in combination with GDC0941 or lapatinib. From the drug screening, we identified as a top candidate vorapaxar, a thrombin-receptor PAR1 inhibitor used as anti-thrombotic agent (Figs 6A and EV5D). Biochemical analysis confirmed the increased phosphorylation of EGFR induced by PAR1 inhibition and a stronger suppression of phospho-AKT and phospho-S6 when vorapaxar was combined with GDC0941 or lapatinib (Fig 6B), phenocopying the phenomenon previously observed in the same cells as a consequence of GNB2 inactivation.

To formally prove that PAR1 could activate AKT through the β subunit of G proteins in these PTEN-null cells, we tested the induction of phospho-AKT after treatment with control or PAR1-agonist peptides in WT and GNB2 KO cells (Fig 6C). While in the WT MDA-MB-468 cells, the agonist was able to increase the phosphorylation

of AKT and ERK compared to control, no activation of these downstream nodes was detected in GNB2 KO cells, demonstrating that GNB2 was necessary for the PAR1 stimulating signal to be

transduced. Also, we confirmed the on-target effect of vorapaxar, since the co-treatment with the drug abrogated the increase in phospho-AKT induced by the agonist peptide in WT cells. We then

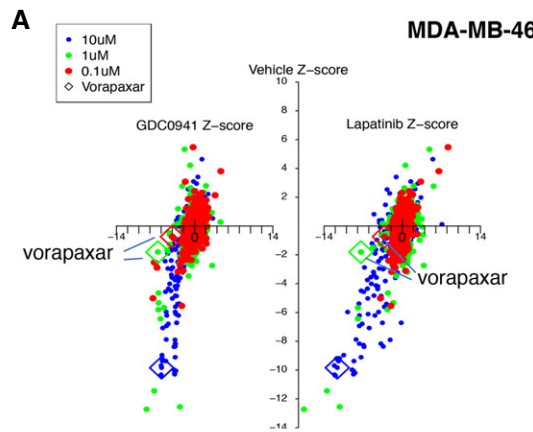

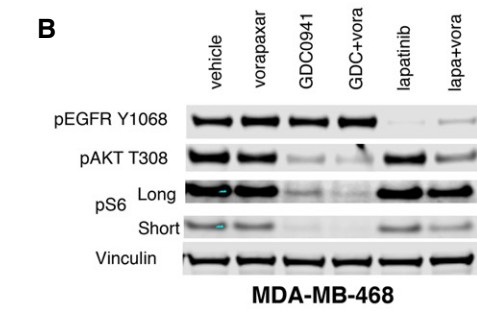

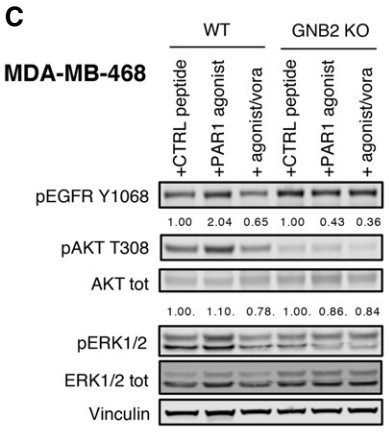

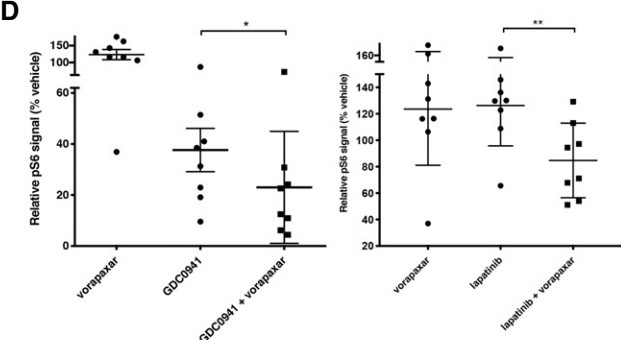

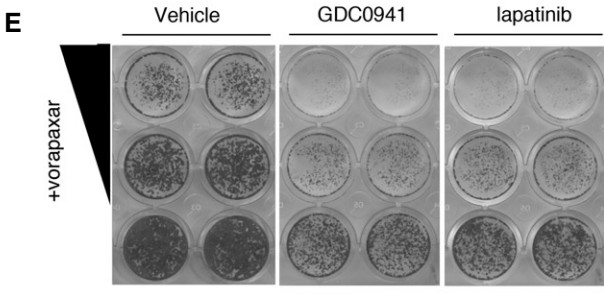

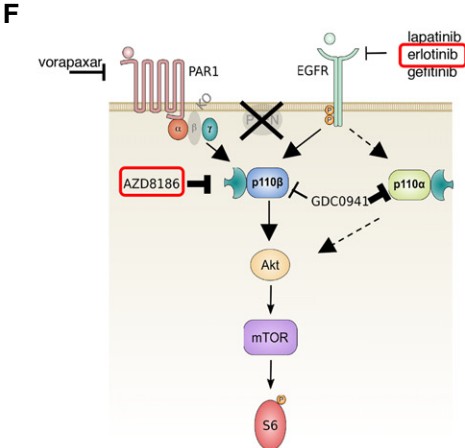

**Figure 6.**

◄

**Figure 6. PAR1 signals through GNB2 to sustain the activation of PI3K pathway in the presence of PI3K or HER inhibitors.**

A   Schematic of the results from the drug screening with compounds targeting GPCR signaling. MDA-MB-468 cells were treated with the compounds of the GPCR-targeted library at three different concentrations (0.1, 1, or 10 μM) in combination with vehicle, GDC0941 450 nM, or lapatinib 1 μM. The pS6 signal was measured by IF after 24 h of treatment and normalized to DAPI. Z scores of the normalized fluorescence values for each drug measured in the presence of vehicle, GDC0941, or lapatinib are reported on the *y*-axis or on the left or right *x*-axis of the dot-plot, respectively. Readings acquired following treatment with the three different concentrations of the GPCR-targeted drugs are reported in three different colors. Dots corresponding to vorapaxar treatments at the three different concentrations are highlighted in the plot. The values reported are mean of a biological triplicate of the experiment.

B   MDA-MB-468 was treated for 24 h with vehicle, vorapaxar (10 μM), GDC0941 (1 μM), or lapatinib (1 μM) alone or in the indicated combinations. The cell lysates were probed with the indicated antibodies.

C   MDA-MB-468 parental cells or GNB2 KO clones were starved and treated with scramble or PAR1 activating peptide for 5 min, alone or in combination with vorapaxar. The cell lysates were probed with the indicated antibodies. Phospho-AKT, pan-AKT, phospho-ERK1/2, and pan-ERK1/2 bands were quantified by the use of ImageJLite software: The ratio of phospho-AKT to pan-AKT and phospho-ERK1/2 to pan-ERK1/2 normalized to the control peptide-treated conditions for WT and GNB2 KO cells (first and fourth lanes, respectively) is shown.

D   p-S6/loading control signal from Western blot experiments in which a panel of six TNBC PTEN-null cell lines and HCC70 cells that acquired resistance to AZD8186 or MK2206 were treated with vehicle, vorapaxar (10 μM), GDC0941 (1 μM), or lapatinib (1 μM) alone or in the indicated combinations. The values were normalized to vehicle treatment for all cell lines. Mean of 2 independent experiments ± SD. *P*-values calculated by two-tailed paired student *t*-test *$P = 0.0235$ and **$P = 0.002$. Cell lines used in the experiments were as follows: MDA-MB-468, HCC70, HCC1937, HCC38, HCC1395, BT-549, HCC70 AZD8186-resistant, and HCC70 MK2206-resistant.

E   Long-term proliferation assay of MDA-MB-468 cells treated with vorapaxar (5 or 2.5 μM), GDC0941 (1 μM), or lapatinib (1 μM), alone or in the indicated combinations. Cells were treated for 2 weeks and stained by crystal violet. One representative experiment of three is shown.

F   Schematic of the network controlling the phosphorylation of AKT and S6 in PTEN-null TNBC cells. EGFR and PAR1-βγ subunits of G protein signal in parallel to p110β and AKT. The two arms of the pathways are inter-connected by feedback compensation mechanisms that act when any of the two signals is perturbed. Combined targeting of both the signaling arms results in a sustained inhibition of the downstream pathway. Erlotinib (EGFR inhibitor) and AZD8186 (p110β inhibitor) are highlighted in the figure as these drugs proved effective in *in vivo* experiments.

Source data are available online for this figure.

extended our findings employing a panel of PTEN-null TNBC cell lines, and we found that the combinatorial treatment of vorapaxar and GDC0941 or lapatinib led to an improved inhibition of the levels of phospho-S6 in multiple cell models (Fig 6D). We also observed a cooperative inhibition in cell viability by the simultaneous treatment of vorapaxar and pan-PI3K or HER inhibitor (Fig 6E).

These data reveal a new element in the signaling cascade that operates in PTEN-null TNBCs and suggest that the GPCR-PAR1 signals upstream of G protein βγ subunits to sustain the activation of AKT.

The results above describe a signaling network in PTEN-deficient TNBC that underlies the sustained activation of the PI3K pathway and relies on two signaling branches. The first upstream activator branch is EGFR family receptor tyrosine kinases, while the second is GPCRs, among which we identified the thrombin receptor PAR1, that in turn activate G proteins, including their βγ subunits. Both axes feed into the activation of PI3Kβ isoform and can compensate each other also through the engagement of different effectors, such as PI3Kα, to overcome the blockade of PI3K pathway and support the phosphorylation of the signaling node AKT and of the downstream marker S6 (Fig 6F).

## Discussion

PTEN deficiency is one of the most common alterations found in human cancers and in triple-negative breast cancers. Given the frequency of the disease in the population (Cancer Research UK statistics 2015 https://www.cancerresearchuk.org/health-professional/cancer-statistics/statistics-by-cancer-type/breast-cancer) and the frequency of PTEN alterations in this type of tumor (Cancer Genome Atlas Network, 2012), we can estimate that up to 15,000 patients may be diagnosed every year in the United States and 22,000 per year in the European Union with a PTEN-deficient triple-negative invasive breast cancer.

PTEN loss results in increased activation of PI3K (Stambolic *et al*, 1998; Haddadi *et al*, 2018) and addiction to PI3K pathway activity (Chen *et al*, 2006; Jia *et al*, 2008; Wee *et al*, 2008; Vasudevan *et al*, 2009; Sangai *et al*, 2012; Hancox *et al*, 2015) in the affected cells. Also, RAS mutations, that are known to confer resistance to a number of therapies (Konieczkowski *et al*, 2018) and to modify the profile of dependencies to PI3K isoform-specific inhibitors (Schmit *et al*, 2014), rarely occur in these tumors. The known biochemical properties of PTEN-deficient cells, both from preclinical models and analysis of the genetic make-up of these tumors from patients, make PI3K pathway inhibition one of the most appealing approaches for the targeting of triple-negative PTEN-null breast cancers.

However, no clear evidence of benefit for PTEN-null cancer patients treated with PI3K inhibitors in clinics has been reported to date (Kim *et al*, 2017; Martin *et al*, 2017) and, despite the urgent need of a targeted treatment for this subset of poor prognosis patients, the molecular bases for the lack of efficacy of those therapies are still poorly understood.

We describe here a signaling network that relies on EGFR and GPCR activity, converges on the activation of PI3Kβ and operates prevalently in PTEN-null breast tumor cells. Inhibition of the GPCR-PI3Kβ axis leads to a rebound in the activity of EGFR and other non-PI3Kβ downstream effectors, such as PI3Kα, to sustain the activation of the pathway. These findings are in line with previous reports showing that the use of combined pan-PI3K class I and EGFR inhibitors produced synergistic responses in basal-like breast cancers (She *et al*, 2016) and that inhibition of PI3Kβ relieves a feedback activation of PI3Kα (Schwartz *et al*, 2015), and they add another piece of information to the complex network that allows PTEN-null breast cancer cells to survive PI3K pathway inhibition. We have translated a deeper knowledge of the signaling into new potential therapeutic strategies, suggesting that PI3Kβ is a good target for inhibition combined with EGFR inhibitor in this tumor type.

EGFR is amplified or overexpressed in a high percentage of triple-negative breast cancers and more frequently than in other

types of breast cancers (Reis-Filho & Tutt, 2008) (Reis-Filho et al, 2006; Reis-Filho & Tutt, 2008; Gumuskaya et al, 2010; Shao et al, 2011; Martin et al, 2012; Park et al, 2014; Nakai et al, 2016). This further emphasizes the importance of this receptor in the biology of these tumors. Given the frequency of its aberrations, EGFR represents an appealing target for inhibition in this type of cancer. However, anti-EGFR therapies tested in clinical trials as monotherapy on triple-negative breast cancers failed to produce beneficial results (von Minckwitz et al, 2005; Dickler et al, 2008; Carey et al, 2012; Yardley et al, 2016). The high prevalence of alterations targeting components of pathways downstream EGFR, such as PTEN, and compensatory feedback signals like the one we report here, may provide a molecular framework to explain such failures.

We have shown that the use of anti-PI3Kβ and anti-EGFR therapies shows an enhanced combination effect in PTEN-null cells. The application of this therapeutic strategy may result in a higher selectivity against the PTEN-deficient tumor cells relative to PTEN wild-type cells and, therefore, in a wider therapeutic window of the treatments compared to the more complete inhibition of different points on the PI3K pathway by a pan-PI3K, AKT, or mTOR inhibitor. This new combinatorial therapy includes drugs that are approved for use in cancer treatment, like anti-EGFR therapies, or that are currently in clinical trials, like PI3Kβ inhibitors, making these findings easier to translate in the clinical practice. Indeed, toxic effects associated with on-target inhibition of PI3Kα, such as hyperglycemia (Bendell et al, 2012), and the related compensatory release of insulin by the pancreas, that in turn can promote resistance to PI3K inhibition by reactivation of insulin signaling in cancer cells (Hopkins et al, 2018), may be limited by the specific targeting of the β isoform of PI3K. Hyperglycemia has not been reported to be among the adverse effects of AZD8186 treatment (Lillian et al, 2016).

In the current study, we used AZD8186 as a PI3Kβ inhibitor. AZD8186 was reported to have higher activity on p110β (IC50 = 4 nM) and p110δ (IC50 = 12 nM), compared to p110α (IC50 = 35 nM) or p110γ (IC50 = 675 nM). We showed that AZD8186 exerted its anti-tumor function in vitro due to PI3Kβ inhibition, since PTEN-null TNBC cells do not express appreciable levels of p110δ and the specific inhibition of p110α in combination with EGFR did not produce the same effects. These results confirmed previous evidence showing AZD8186 to selectively target the β over the α isoform of PI3K in breast cancer cells (Schwartz et al, 2015). We cannot formally rule out the possibility that AZD8186 may have some function in vivo related to inhibition of p110δ, especially in immune cells, that are known to express high levels of this protein. However, the likelihood that the response observed in vivo may be due to interference with the immune-compartment by AZD8186 is low, since results have been confirmed in different models and in both immune-competent and immune-deficient recipients.

We report here that GPCRs, and in particular PAR1, are responsible for the activation of PI3Kβ and for the sustained activity of PI3K pathway also in the presence of EGFR inhibitors. PAR1 is one of the four members of the protease-activated receptors (PARs) GPCR family and is activated by thrombin and by a variety of other tumor-associated proteases, including plasmin and MMP-1 (Boire et al, 2005). Indeed, PAR1 expression has been positively correlated with carcinoma cell invasiveness (Even-Ram et al, 1998) and its altered trafficking and persistent signaling was shown to promote breast cancer invasion (Booden et al, 2004).

PAR1 and EGFR pathways have been previously functionally linked. PAR1 was reported to transactivate EGFR not only in vascular smooth muscle cells (Kanda et al, 2001) and in cardiac fibroblasts (Sabri et al, 2002), but also in colorectal (Darmoul et al, 2004) and in breast cancers (Arora et al, 2008) through a variety of different mechanisms, showing that the two pathways can reciprocally influence their activities. Also, a role for PI3Kβ in mediating the synergistic production of $PIP_3$ and activation of AKT in response to both GPCR and RTK inputs has been shown (Kurosu et al, 1997; Murga et al, 2000; Ciraolo et al, 2008; Hauser et al, 2017) and this provides a molecular framework to our observation that EGFR and PAR1 can simultaneously signal through PI3Kβ in triple-negative breast cancers. Our data describe how those two branches of the pathway can compensate each other following PI3K inhibition and place this GPCR-RTK signaling network at the centre of a mechanism of drug resistance. The stronger suppression of basal PI3K pathway activity induced by GNB2 inactivation compared to PAR1 inhibition suggests the possibility that other GPCRs in addition to PAR1 may contribute to the signaling to PI3Kβ in these cells. Among other GPCRs that might be involved in this pathway, GPER1 could be a possible candidate, as it was previously shown to mediate signaling from 17beta-estradiol to βγ subunits of G protein and transactivation of EGFR in ER-negative breast cancers (Filardo et al, 2000).

However, our data demonstrate a role of PAR1 in mediating resistance to PI3K and EGFR inhibitors and suggest that GPCRs may represent important drivers of drug resistance also in other contexts. Since GPCR drug discovery has been an area of intense activity (Hauser et al, 2017), we envision that these findings may have significant potential for translation.

In addition to implicating GPCR signaling, our study also highlighted an unexpected role of CK2 inhibition in potentiating the activity of PI3K pathway inhibitors in PTEN-null TNBC cells. Previous work showed that PTEN deletion affects levels of CK2 through transcriptional STAT3-mediated upregulation (Kalathur et al, 2015), suggesting that this protein may have an important role in PTEN-null tumor cells. CK2 is in turn known to phosphorylate AKT in Ser129 and to contribute to its activation (Di Maira et al, 2005), providing a molecular framework to its role in limiting response to PI3K pathway inhibitors. It has also been reported that suppression of PI3K-AKT-mTOR pathway was enhanced by combined targeting of EGFR and CK2 in lung cancer models relying on EGFR activity (Bliesath et al, 2012). CK2 could thus expand the array of potential drug targets for the design of combinatorial treatments in PTEN-null TNBCs beyond EGFR and PI3Kβ.

# Materials and Methods

### Cell lines

MDA-MB-468, BT-549, U-251, A172, U-87-MG, BT-20, MDA-MB-231, and MDA-MB-361 were maintained in DMEM supplemented with 10% FBS. MCF7 were maintained in DMEM supplemented with 10% FBS and Insulin Human Solution 1:1,000 (I9278 Sigma Aldrich). T47D were maintained in RPMI with 10% FBS and Insulin Human Solution 1:1,000. Cell lines derived from mouse primary mammary gland tumors were established in medium consisting of DMEM/F12 Glutamax (Thermofisher), 10% FBS

heat-inactivated, 1:1,000 dilution of Insulin solution, and EGF 20 µg/ml, and they were cultivated on collagen solution-coated plates. After 10 passages, cells were adapted to grow in DMEM 10% FBS and on normal tissue culture plates. HCC70 and ZR-75-1 acquired resistance models were maintained in RPMI supplemented with 10% FBS and AZD8186 250 nM or MK2206 1 µM. The rest of the cell lines were cultured in RPMI with 10% FBS. Cell lines were tested for mycoplasma and were authenticated by short-tandem repeat (STR) DNA profiling by the Francis Crick Institute Cell Services facility.

## Compounds and reagents

AZD8186 was obtained from a collaboration with AstraZeneca. Gefitinib, erlotinib, lapatinib, GDC0941, MK2206, and vorapaxar were purchased from Selleckchem. CX-4945 was obtained from MedChemExpress. Cetuximab was a kind gift from Charles Swanton.

The library of compound targeting GPCR signaling was purchased from MedChemExpress (HY-L006).

Antibodies for immunoblots purchased from Cell Signaling Technology were as follows: anti-phospho-EGFR Y1068 (Cat# 3777, dilution 1:1,000), EGFR (Cat# 4267, dilution 1:1,000), phospho-AKT T308 (Cat# 13038, dilution 1:1,000), S473 (Cat# 9271, dilution 1:1,000) and S129(Cat#13461, dilution 1:1,000), AKT (Cat# 2920, dilution 1:1,000), phospho-ERK T202/Y204 (Cat# 9101, dilution 1:1,000), ERK (Cat# 9107, dilution 1:2,000), phospho-S6 S235/236 (Cat# 2211, dilution 1:4,000), S6 (Cat# 2317, dilution 1:500), phospho-PRAS40 T246 (Cat# 2640, dilution 1:1,000), PTEN (Cat#9559, dilution 1:1,000), p110α (Cat# 4249, dilution 1:400), and p110β (Cat# 3011, dilution 1:1,000). Anti-vinculin was from Sigma-Aldrich (Cat# V4505, dilution 1:5,000). Anti-GNB2 (ab81272, dilution 1:1,000) was from Abcam and anti-Sos1 from Santa Cruz Biotechnology (sc-17793, dilution 1:1,000). Anti-EGFR1 (from Francis Crick Institute Cell Service) was used in the Immuno-precipitation experiments (5 µg antibody/1 mg protein).

pLenti_BSD_sgRNA plasmid was a generous gift of Paola Scaffidi, and it was generated through replacement of GFP cassette with BSD cassette into the pLenti-sgRNA-Lib from Wei lab (Addgene Plasmid #53121). pCW-Cas9 vector generated in David Sabatini's lab was obtained from Addgene (#50661).

## In vivo studies

All studies were performed under a UK Home office approved project license and in accordance with institutional welfare guidelines.

Wap-cre:Pten$^{fl/fl}$:Tp53$^{fl/fl}$ mouse model was generated crossing the Trp53$^{tm1Brn/tm1Brn}$ (NCI Mouse Repository) and the Pten$^{tm1Hwu/tm1Hwu}$ lines, previously back-crossed in C57BL/6 background, with the Wap-cre strain generated in C57BL/6 background by the NCI Mouse Repository.

Female mice were allowed to breed and wean their pups in order to activate the expression of Cre transgene. After that, mice were monitored for mammary tumor growth. Tumors were measured using caliper, and volume was estimated using the formula width$^2$ × length × 0.5. Before the tumors reached the size-limit imposed by the project license, mice were culled and the mammary tumor extracted.

For human cell line *in vivo* studies, 5 million cells were re-suspended in PBS mixed 1:1 with growth-factor-reduced Matrigel and injected into the fat pad of the left, fourth mammary gland of 6- to 8-week-old female NU(NCr)-Foxn1$^{nu}$ (Charles River). Tumor volumes were determined using the formula width$^2$ × length × 0.5. When tumors reached a volume of 100 or 250 mm$^3$, mice were randomly assigned to treatment with vehicle or drugs.

For spontaneous tumor-derived cell lines, 5–10 millions of cells were re-suspended in 1:1 PBS: growth-factor-reduced Matrigel and injected into the fat pad of the left, fourth mammary gland of 6- to 8-week-old female C57BL/6 mice.

For *in vivo* drug treatments, AZD8186 was formulated in 0.5% (hydroxypropyl) methyl-cellulose/0.2% Tween-80 and administered by oral gavage (5 µl/g) every day. Erlotinib was prepared in 0.3% (hydroxypropyl) methyl-cellulose and administered intra-peritoneum every day at 5 µl/g.

## shRNA screening

Whole-genome shRNA was performed using the MISSION LentiPlex Pooled shRNA library from Sigma (SHPH01). The library contains 80717 shRNA constructs from the TRC collection targeting around 16,000 genes and is divided into 10 different pools, which were infected and sequenced separately.

MDA-MB-468 cells were infected in triplicate with the ready-to-use lentivirus in the presence of 8 µg/ml polybrene at a multiplicity of infection MOI = 0.8 and at an initial representation of 800 cells for each shRNA. Forty-eight hours after the infection, cells were selected in 2 µg/ml puromycin for 48 h, followed by 24-h growth with fresh media. Then, cells were divided into five different aliquots. One aliquot was frozen to measure the initial shRNA representation (time = 0). The other four aliquots were seeded in 15-cm dishes, and the day after cells were treated with either DMSO, 250 nM AZD8186, 250 nM GDC0941, or 450 nM MK2206. Drugs were replaced every 3 days of treatment and cells passaged after 6 days treatment keeping a representation of 400. After another cycle of 6 days of treatment, cells were trypsinized, counted to measure the efficiency of the treatment, and frozen. Also the genomic DNA extraction and the PCR detailed below were done in order to keep a 400 representation of each shRNA.

Genomic DNA was isolated from all the samples using Wizard genomic DNA purification kit (Promega). shRNA inserts were retrieved from the genomic DNA by PCR amplification using the following conditions: (i) 98°C, 30 s; (ii) 98°C, 10 s; (iii) 60°C, 20 s; (iv) 72°C, 1 min; (v) to step 2, 16, or 11 cycles (for PCR1 or PCR2, respectively); 72°C, 5 min. Indexes and adaptors for deep sequencing (Illumina) were incorporated in the PCR primers. For PCR1, gDNA was amplified using Illuseq_x_PLKO1_f ACACTC TTTCCCTACACGACGCTCTTCCGATCTxxxxxx CTTGTGGAAAGGA CGAAACACCGG (where x indicates different barcodes) and P7_pLKO1_r CAAGCAGAAGACGGCATACGAGATTTCTTTCCCCTG CACTGTACCC primers. 2.5 µl of PCR1 product was used as templates for PCR2 reaction, together with P5_IlluSeq AATGATA CGGCGACCACCGAGATCTACACTCTTTCCCTACACGACGCTCTTCC GATCT and P7 CAAGCAGAAGACGGCATACGAGAT primers. Final PCR product was purified using MiniElute PCR Cleanup (Qiagen) and quantified using Bioanalyzer. Finally, shRNA representation for each sample was measured by next-generation sequencing

(Illumina). The shRNA sequences were extracted from the sequencing reads and aligned to TRC library. Due to the customized vector structure, samples pooled into the same sequencing lane could not be demultiplexed with the standard Illumina pipeline. As such, in-house software was used to recognize the first six nucleotides of each read as the barcode, and nucleotides 31–53 as the shRNA sequence. Each sample was then collapsed to "tags", so that each unique sequence was represented in the file only once, along with a count of the total number of times it was detected. These tag files were then mapped to the TRC library sequences using bwa 0.5.9 (Li & Durbin, 2009) with –n 4, -k 4, -l 90 (i.e., seed length > read lengths and maximum mismatches = 4). Counts of the number of times each sequence within the targeted pools was detected (taking into account the count associated with each tag) were then summarized with off-target reads discarded. shRNAs not represented by at least 50 reads in each sample were removed at this stage. Counts were then normalized to the maximum total number of aligned reads across all samples for that pool. Comparisons were performed by considering the mean fold change across triplicates, with $P$-value calculated by a paired $t$-test on these log values. The following criteria were applied to filter genes. At least one shRNA associated with a mean ratio of absolute counts between drug treatment and vehicle $\leq 0.65$ with a $t$-test $P$-value between triplicates $\leq 0.15$; at least one shRNA associated with a mean ratio of absolute counts between drug treatment and vehicle $\leq 0.75$ with a $t$-test $P$-value between triplicates $\leq 0.15$; and mean ratio of absolute counts between vehicle and time 0 > 0.5. Filtered genes were then ranked based on the lowest mean ratio of absolute counts between drug treatment and vehicle for the shRNA associated with the second lowest ratio for each gene (Table EV1).

## CRISPR-Cas9 library generation

Ninety-six candidates were selected among the 163 genes for which at least two different shRNAs showed mean ratio of absolute counts treatment/vehicle $\leq 0.5$ in the shRNA screening (Table EV1).

In addition, we included genes codifying for epitopes that were down- or up-regulated by AZD8186 treatment in the RPPA analysis by a $Log_2$ Fold change $\geq 0.25$ and with a $P$-value $\leq 0.05$ at any time point.

Based on biological interest, we also included some genes at the edge of inclusion criteria (reported in Table EV2 not in bold), PIK3CA (as a positive control based on previous publications) and we excluded AKT, S6, NDRG1, and PRAS40.

Four or five not-overlapping sgRNA sequences were selected for each of those genes from the list reported in Wang *et al* (2015) and synthesized (Sigma-Aldrich) with forward strand 5′ overhang -ACCG and reverse strand 5′ overhang –AAAC. Forward and reverse strands of the oligos were annealed and ligated by Golden Gate Assembly into the pLenti_BSD_sgRNA plasmid, as previously described (Zhou *et al*, 2014), with minor modifications. Briefly, the annealing was performed incubating forward and reverse primers at 10 μM each (final concentration) in PNK 1× buffer at 95°C for 5 min, and then, the temperature was ramped to 25°C at 0.1°C per second. Two microliter of the annealed primer solution diluted 1:200 was used in the Golden Gate assembly reaction together with BsmBI (five units), pLenti_BSD_sgRNA (50 ng), T7 DNA ligase (1,500 units), T4 ligase

buffer with ATP (diluted to 1×) up to 10 μl final volume for each reaction. The following conditions were used as follows: 25 cycles of 45°C for 2 min and 20°C for 2 min, then 60°C for 10 min, and 80°C for 10 min.

Ligated vectors were transformed by heat-shock in Stbl3 competent E. Coli bacteria (one vector in each well of 96-multi-well plates). Bacteria were grown overnight in 96-deep-well plates, and after growth, bacteria transformed by sgRNA vectors targeting the same gene were pooled together before plasmid extraction (Qiagen QIAprep Spin Miniprep Columns). Lentiviral particles were produced in arrayed format by transfecting HEK293T cells in 12-well tissue culture plates with 67.6 ng pCMV-VSVG, 203 ng pCMV-8.2 (Addgene), and 270 ng pLenti_sgRNA vector mixture for each gene in separated wells (Lipofectamine, Invitrogen). Forty-eight hours after transfection, virus particles in the supernatant were harvested and stored at −80°C in 96 well plates.

## CRISPR-Cas9 KO screening

An MDA-MB-468 clone expressing a DOX-inducible form of Cas9 was derived by transduction of the parental cells with pCW-Cas9 lentivirus. Infected cells were selected by hygromycin B (Thermo-Fisher Scientific, final concentration 200 μg/ml) and single-cell cloned in 96-well tissue culture plates.

MDA-MB-468 iCas9-cloned cells were seeded at 10,000 cells/well in 24-well plates and infected in array format with the pLenti_BSD_sgRNA library in the presence of polybrene 8 μg/ml (200 μl viral supernatant/well). Cells were selected with blasticidin S for 5 days (ThermoFisher Scientific, final concentration 5 μg/ml). After selection, cell numbers were extrapolated by counting cells in three wells and by comparing their Cell Titer Blue Viability assay (Promega) readings with the readings of all other samples. Cells were seeded at 50,000 cells/well in 12-well plates and treated with 1 μg/ml doxycycline for 4 days. After that, cells were counted as previously described and passaged in equal numbers in wells of similar size before being treated for 4 days by doxycycline. Cells were counted again and seeded at a density of 4,000/wells in 96-well black plates before being treated with vehicle, AZD8186 100 nM, GDC0941 400 nM, or MK2206 450 nM in technical quadruplicate. After 30 h of treatment, cells were fixed in 96-well black plates by 4% formaldehyde (15 min of incubation), washed three times in PBS, blocked 1 h in 1× PBS/5% normal serum (Cell Signaling Technology)/0.3% Triton X-100 (Sigma-Aldrich), and incubated overnight in antibody dilution solution (1× PBS/1% BSA/0.3% Triton™ X-100) containing anti-phospho S6 S240/244 antibody (Cell Signaling Technology Cat#5364) diluted 1:1,200. Plates were then washed three times in PBS and incubated for 1 h in antibody dilution solution containing Anti-rabbit IgG Alexa Fluor-488 Conjugate (Cell Signaling Technology #4412) diluted 1:1,000 and DAPI 0.2 μg/ml. Plates were washed three times in PBS, and images were acquired and signal quantified using CX7 LZR high content microscope (ThermoFisher Scientific). The fluorescent signal from each well was normalized to DAPI.

All washing steps of 96-well plates were performed by 96-well head Biomek FX liquid handling robot (Beckman).

The screening experiment previously described was performed in triplicate.

For analysis, per-sample data were scaled to the maximum sum across samples (analogous to normalizing count data to the maximum yield across samples), $\log_2$ transformed and $P$-values are associated with a two-sided $t$-test. Differences were considered significant if the $P$-value remained $< 0.05$ after Bonferroni correction.

### GNB2 knockout

MDA-MB-468 iCas9 cells were transduced with four sgRNAs targeting GNB2 from the pLenti_BSD_sgRNA library, selected by Blasticidin S, and treated with doxycycline as previously described. Cells were then cloned in 96-well plates, and clones were screened for the lack of GNB2 protein by immunofluorescence in black 96-well plates with the same procedure described in the CRISPR Cas9 KO screening and using anti-GNB2 primary antibody (ab81272).

### GPCR-centered drug screening

Drug screening was performed in triplicate using three different concentrations (0.1, 1, and 10 µM) for each of the 716 drugs included in the library MedChemExpress (HY-L006). Echo550 acoustic dispenser (Labcyte) within an Access robotic workstation with GX robotic arm was employed to add compounds at the right concentrations to 384-well plates previously seeded with 900 cells/well. After 24-h treatment, cells were fixed, washed, and stained by anti-phospho S6 S240/244 antibody (Cell Signaling Technology Cat#5364) and DAPI as previously described. Images were acquired and quantified as previously described. The fluorescent signal in each well was normalized to DAPI and expressed as $z$-score in each plate.

### Reverse-phase protein array

MDA-MB-468 cells were seeded at 350,000 cells/well in six-well plate. After cells were attached, they were treated with vehicle or AZD8186 250 nM for 2 or 28 h. Cell lysates were obtained by the recommended lysis buffer (https://www.mdanderson.org/research/research-resources/core-facilities/functional-proteomics-rppa-core.html), and cell lysates were submitted to MD Anderson Cancer Center RPPA Core Facility. Linear normalized data were analyzed by predicting the mean expression level using a linear model for each time point, with drug status, target and their interaction term as predictors. $P$-values associated with the test for a nonzero coefficient of the interaction term were calculated, and the results were plotted as the mean target-specific drug effect against the $P$-value.

### Proliferation assays

Cells were seeded in 96-well plates, and drugs were added on the following day. For 6-day viability assays, drugs were replaced after 3 days of treatment. Cell viability was determined by Cell Titer Blue Viability Assay (Promega).

For long-term assays, cells were seeded into 24-well plates and treated with drugs for 14 days. Drugs were replaced every 3 or 4 days. Cells were fixed and stained with a solution containing 2% ethanol and 0.2% crystal violet. In both assays, starting cell density

### The paper explained

#### Problem

PI3k pathway inhibitors (PI3Kpi) represent a rational choice for the treatment of triple-negative breast cancers (TNBC) that, as a consequence of loss of PTEN tumor suppressor function, show aberrant PI3K pathway activation. Indeed, pre-clinical models of these tumors showed sensitivity to PI3Kpi and especially to inhibition of PI3Kβ. However, clinical efficacy of these drugs has been so far modest, setting the need for more efficient combinatorial approaches.

#### Results

Through a combination of unbiased screenings and hypothesis-driven approaches, we identified a molecular network that impairs response to PI3Kpi in PTEN-null TNBCs. Both the G protein-coupled receptor PAR1, through engagement of the βγ subunit of G protein, and EGFR were discovered to signal to PI3Kβ in these tumors. The two branches of this pathway can compensate each other and promote the sustained activation of AKT to override PI3Kpi-mediated blockade. Simultaneous inhibition of PI3Kβ and EGFR efficiently blunted the activation of the pathway and produced anti-tumor activity both *in vitro* and *in vivo* in different PTEN-null TNBC models.

#### Impact

This study unveiled signaling nodes that are fundamental for the survival of PTEN-null TNBCs in the presence of PI3K pathway inhibitors. It also highlighted the combinatorial targeting of PI3Kβ and EGFR as a potential therapeutic strategy to meet the clinical need of treating PTEN-null TNBCs.

was optimized to produce an 80–90% confluent monolayer in vehicle-treated cells at the conclusion of the experiment.

### Western blot

Protein cell lysates were extracted by boiled Laemmli buffer (2.5% SDS, 125 mM Tris–HCl, PH 6.8), and the lysates were sonicated. The protein concentration of the supernatant was determined by the micro-BCA protein assay (Pierce). Equal amounts of whole cell lysate per lane were boiled in LDS buffer and reducing agent, according to the manufacturer instructions, and separated by electrophoresis in 4–12% gradient NuPAGE Novex Bis-Tris gels (Life Technologies) under reducing conditions, and subsequently transferred to polyvinylidene difluoride membranes (Millipore Immobilon-P). Bound primary antibodies were incubated with horseradish peroxidase-conjugated secondary antibodies and detected using chemiluminescence (Luminata HRP substrate, Millipore). Alternatively, membranes were incubated with secondary conjugates compatible with infrared detection at 700 and 800 nm, and membranes were scanned using the Odyssey Infrared Imaging System (Odyssey, LICOR). Western blot quantification was done using ImageStudioLite software.

### Immunoprecipitation

Protein cell lysates were extracted by NP-40 lysis buffer (100 mM TrisCl, pH8.3, 100 mM NaCl, 0.5% Nonidet P-40—Roche) supplemented with tablets of protease and phosphatase inhibitor cocktails (Roche). Cell lysates were cleared by centrifugation at 4°C for

10 min, and the supernatants were collected and quantified by micro-BCA protein assay (Pierce). One milligram of protein lysates was pre-cleared by 50 µl of protein G-Sepharose beads (GE Healthcare 17-0618-01) slurry/sample rotating 30 min at 4°C. Pre-cleared lysates were centrifuged, and the supernatant was incubated rotating overnight with EGFR1 antibody (see compounds and reagents section). The immune-precipitates were then incubated 2 h at 4°C with 50 µl of Agarose-Ig beads, washed five times with washing buffer (50 mM TrisCl, pH8.3, 150 mM NaCl, 0.05% Nonidet P-40 supplemented with protease and phosphatase inhibitor tablets) and re-suspended in LDS buffer and reducing agent. The immune-complexes were the separated by electrophoresis and transferred on polyvinylidene difluoride membranes as previously described and probed with the indicated antibodies.

### Quantitative RT–PCR

RNA was isolated (Qiagen), and reverse transcription was conducted (Applied Biosystems) using standard methods. Quantitative real-time PCR was conducted using gene-specific primers (QuantiTect Primer Assays, Qiagen) for EGFR and CSNK2B with Fast SYBR Green Master Mix (Applied Biosystems).

## Data availability

Data from shRNA screen are deposited in GEO GSE148785 (https://www.ncbi.nlm.nih.gov/geo/query/acc.cgi?acc = GSE148785).
Source data from the CRISPR-Cas9 KO screen are reported in Table EV3. Data from the GPCR-centered drug screen are reported in Table EV4.

**Expanded View** for this article is available online.

## Acknowledgements

We thank Miriam Molina and David Hancock for helpful discussions and critical reading of the manuscript, Romain Baer for graphical representations of the signaling pathway and the science technology platforms at the Francis Crick Institute including Biological Resources, Advanced Sequencing Facility, Computational Biology, Genomics Equipment Park, Experimental Histopathology and Cell Services. This work was supported by funding to JD from the Francis Crick Institute, which receives its core funding from Cancer Research UK (FC001070), the UK Medical Research Council (FC001070), and the Wellcome Trust (FC001070), from the European Research Council Advanced Grant RASIMMUNE and from a Wellcome Trust Senior Investigator Award 103799/Z/14/Z. DZ was recipient of a post-doctoral fellowship from the Fondazione Umberto Veronesi—Young Investigator Programme 2013.

## Author contributions

DZ and JD designed the study, interpreted the results, and wrote the manuscript. DZ and FM performed the biochemical experiments, CM assisted with *in vivo* studies, SH performed bioinformatics analyses, SR assisted with viability experiments, MH provided expertise and carried out screening experiments. All authors contributed to manuscript revision and review.

## Conflict of interest

The authors report no conflicts of interest related to this work.

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
