## [Review Process File · EMBO Molecular Medicine]

Targeting of GPCR and EGF receptor overcomes PI3K inhibitor resistance in PTEN-null breast cancer

Davide Zecchin, Chris Moore, Fanis Michailidis, Stuart Horswell, Sareena Rana, Michael Howell, and Julian Downward

DOI: [10.15252/emmm.202011987](https://doi.org/10.15252/emmm.202011987)

Corresponding author: Julian Downward (julian.downward@crick.ac.uk)

Review Timeline:

Submission Date:	6th Jan 20
Editorial Decision:	31st Jan 20
Revision Received:	23rd Apr 20
Editorial Decision:	16th May 20
Revision Received:	9th Jun 20
Accepted:	16th Jun 20

Editor: Lise Roth

Transaction Report:

31st Jan 2020

Dear Dr. Downward,

Thank you for the submission of your manuscript to EMBO Molecular Medicine. We have now received feedback from the three reviewers who agreed to evaluate your manuscript. As you will see from the reports below, the referees acknowledge the interest of the study and are overall supporting publication of your work pending appropriate revisions.

Addressing the reviewers' concerns in full will be necessary for further considering the manuscript in our journal, and acceptance of the manuscript will entail a second round of review. EMBO Molecular Medicine encourages a single round of revision only and therefore, acceptance or rejection of the manuscript will depend on the completeness of your responses included in the next, final version of the manuscript. For this reason, and to save you from any frustrations in the end, I would strongly advise against returning an incomplete revision.

When submitting your revised manuscript, please carefully review the instructions that follow below. Failure to include requested items will delay the evaluation of your revision:

2) Individual production quality figure files as .eps, .tif, .jpg (one file per figure).

3) A .docx formatted letter INCLUDING the reviewers' reports and your detailed point-by-point responses to their comments. As part of the EMBO Press transparent editorial process, the point-by-point response is part of the Review Process File (RPF), which will be published alongside your paper.

4) A complete author checklist, which you can download from our author guidelines (<https://www.embopress.org/page/journal/17574684/authorguide#submissionofrevisions>). Please insert information in the checklist that is also reflected in the manuscript. The completed author checklist will also be part of the RPF.

5) Before submitting your revision, primary datasets produced in this study need to be deposited in an appropriate public database (see <https://www.embopress.org/page/journal/17574684/authorguide#dataavailability>). Please remember to provide a reviewer password if the datasets are not yet public. The accession numbers and database should be listed in a formal "Data Availability" section (placed after Materials & Method). Please note that the Data Availability Section is restricted to new primary data that are part of this study.

6) We would also encourage you to include the source data for figure panels that show essential data. Numerical data should be provided as individual .xls or .csv files (including a tab describing the data). For blots or microscopy, uncropped images should be submitted (using a zip archive if multiple images need to be supplied for one panel). Additional information on source data and instruction on how to label the files are available at

7) Our journal encourages inclusion of *data citations in the reference list* to directly cite datasets that were re-used and obtained from public databases. Data citations in the article text are distinct from normal bibliographical citations and should directly link to the database records from which the data can be accessed. In the main text, data citations are formatted as follows: "Data ref: Smith et al, 2001" or "Data ref: NCBI Sequence Read Archive PRJNA342805, 2017". In the Reference list, data citations must be labeled with "[DATASET]". A data reference must provide the database name, accession number/identifiers and a resolvable link to the landing page from which the data can be accessed at the end of the reference. Further instructions are available at .

8) We replaced Supplementary Information with Expanded View (EV) Figures and Tables that are collapsible/expandable online. A maximum of 5 EV Figures can be typeset. EV Figures should be cited as 'Figure EV1, Figure EV2" etc... in the text and their respective legends should be included in the main text after the legends of regular figures.

- Additional Tables/Datasets should be labeled and referred to as Table EV1, Dataset EV1, etc. Legends have to be provided in a separate tab in case of .xls files. Alternatively, the legend can be supplied as a separate text file (README) and zipped together with the Table/Dataset file. See detailed instructions here:

9) The paper explained: EMBO Molecular Medicine articles are accompanied by a summary of the articles to emphasize the major findings in the paper and their medical implications for the non-specialist reader. Please provide a draft summary of your article highlighting

10) For more information: There is space at the end of each article to list relevant web links for further consultation by our readers. Could you identify some relevant ones and provide such information as well? Some examples are patient associations, relevant databases, OMIM/proteins/genes links, author's websites, etc...

11) Every published paper now includes a 'Synopsis' to further enhance discoverability. Synopses are displayed on the journal webpage and are freely accessible to all readers. They include a short stand first (maximum of 300 characters, including space) as well as 2-5 one-sentences bullet points

that summarizes the paper. Please write the bullet points to summarize the key NEW findings. They should be designed to be complementary to the abstract - i.e. not repeat the same text. We encourage inclusion of key acronyms and quantitative information (maximum of 30 words / bullet point). Please use the passive voice. Please attach these in a separate file or send them by email, we will incorporate them accordingly.

Please also suggest a striking image or visual abstract to illustrate your article as a jpeg file 550 px-wide x 400-px high.

15) As part of the EMBO Publications transparent editorial process initiative (see our Editorial at <http://embomolmed.embopress.org/content/2/9/329>), EMBO Molecular Medicine will publish online a Review Process File (RPF) to accompany accepted manuscripts.

In the event of acceptance, this file will be published in conjunction with your paper and will include the anonymous referee reports, your point-by-point response and all pertinent correspondence relating to the manuscript. Let us know whether you agree with the publication of the RPF and as here, if you want to remove or not any figures from it prior to publication.

I look forward to receiving your revised manuscript.

Yours sincerely,

Lise Roth

Lise Roth, PhD
Editor
EMBO Molecular Medicine

To submit your manuscript, please follow this link:

Link Not Available

***** Reviewer's comments *****

Referee #1 (Remarks for Author):

The manuscript by Zecchin et al describes the discovery of a number of potential additional targets that would be useful in PI3K inhibition in PTEN deficient triple negative breast cancer. Intriguingly, the authors find key role of GPCRs and EGFR, which in the context of the PTEN deficiency suggested a targetable strategy of inhibiting the beta isoform of PI3K beta in combination with these identified targets. This research provides an intriguing mechanism to develop novel therapeutic targeted combination strategies towards PTEN deficient TNBC. Overall, this research appears to be well described and would provide a useful advance to researchers both in the PI3K and TNBC fields.

This research is another excellent example of the ability of PI3K beta to synergize inputs from both RTK and GPCR inputs, and provides an exciting new rationale for designing novel combinational therapies.

My only true concern is in the true isoform selectivity of the AZD8186 compound, which should be able to be addressed in a very simple set of experiments.

Major points

1. In the results text the AZD8186 inhibitor is described as beta isoform specific. The IC50 values of this compound have been reported (see below), and calling it beta specific might be a stretch. It inhibits delta with very similar IC50 values (although this isoform would likely not be expressed in these cells), with it being roughly 10 fold selective over alpha.

This could likely be reworded to more clearly describe its specificity, as readers of this manuscript may not catch this subtlety. The authors do note that they use a concentration in cells that will mainly target p110 beta derived from Schwartz et al.

<https://mct.aacrjournals.org/content/14/1/48.long>

<https://www.sciencedirect.com/science/article/pii/S1535610814004590?via%3Dihub>

In this case the authors of that 2015 manuscript determined that AZD8186 had no effect at up to 250 nM in a HER2 p110alpha dependent tumor, so at this concentration it is mainly exerting its effect on p110 beta. This strikes me that this will be cell type dependent based on the actual amount of p110 alpha vs p110 beta that is expressed in the given cell type. These cells likely will not have delta expression, so the determination of the role of p110 alpha vs beta is critical.

It would be useful in one of these cell models (the simplest might be to repeat the work in Fig 1G) to use in addition a p110 alpha specific inhibitor in combination with an EGFR inhibitor to verify the importance of p110 beta. This does not need to be repeated across all cell types, but would reinforce the cited work by Schwartz et al, and support the claims of the important role of p110 beta. The authors have already used this compound in the Fig 5 in the GNB2 knockout.

However, in the mouse experiments (Fig. 2) it is critical to note that the AZD8186 compound will target delta almost as effectively as beta (4 vs 12 nM). While I do not think it is necessary to repeat any of the mouse studies with a delta inhibitor, it is essential that they at least clearly describe this potential complication of the experiment in the results and discussion, as delta inhibition can have other anti-tumor effects.

2. The discovery of the CK2 kinase role in this process is intriguing and unexpected. Can the authors expand in the discussion on the possible role of this kinase in the pathway in the discussion. Currently this is brought up in the results, and not explored in depth in the discussion.

3. There are a number of comments in the discussion that state 'for the first time ...'. I am not sure these novelty phrases improve the work, and I think they should be removed.

Referee #2 (Comments on Novelty/Model System for Author):

This well-written and structured manuscript represents an impressive series of experiments which highlight the importance of two signaling branches, EGFR and GPCR, in PTEN-deficient TNBC using relevant models, and which may have clinical relevance in the future.

Referee #2 (Remarks for Author):

Zecchin et al have performed a whole genome shRNA screen in a PTEN-null TNBC model to identify targets whose inhibition synergizes with inhibitors of PI3K pathway components, namely AZD8186 (inhibiting p110beta), GDC0941 (pan-PI3K), and MK2206 (AKT), followed by a comprehensive series of mechanistic studies in relevant models and validation work. This well-written and structured manuscript represents an impressive series of experiments which highlight the importance of two signaling branches, EGFR and GPCR, in PTEN-deficient TNBC using relevant models, and which may have clinical relevance in the future. Among the shRNA screen hits, the authors focused on EGFR, and the casein kinase 2 components CSNK2B and CSNK2A2, which were selectively depleted in the AZD8186 condition compared to untreated control. Combination treatment of cells with EGFR-inhibitors and PI3K pathway inhibitors was synergistic, as was the combination of EGFR-inhibitors and PI3Kbeta inhibition in human xenografts and in an immunocompetent conditional Pten/p53-null mouse model and isografts. To support clinical relevance, the authors also interrogated the METABRIC database and found EGFR to be overexpressed in PTEN-low/mut breast cancers. To further dissect the signaling downstream of combination treatments, western blotting experiments across PTEN-null cell lines and treatment conditions identified decreased phospho-S6 as a good biochemical readout of response. Co-IP experiments under various treatments showed p110beta to interact with EGFR, which was disrupted most potently by combined p110beta and EGFR inhibition, and that AKT phosphorylation downstream of EGFR signaling was mediated in part by p110beta. Furthermore, to validate the screen and identify additional modifiers, 110 of the shRNA hits plus 31 hits from RPPA analysis of MDA-468 cells treated with AZD8186, were used in a CRISPR-Cas9 screen using pS6 as the readout. The shRNA screen was largely validated; and the authors identified knock out of GNB2 and GNG5 (encoding G protein beta and gamma subunits) as cooperative partners of GDC0941 pan-PI3K inhibition. KO clones of GNB2 had increased EGFR expression, phosphorylation, and proportion of phospho-EGFR, and decreased pAKT and pS6 which was accentuated with GDC0941 treatment. EGFR-inhibition in the GNB2 KO context also led to greater suppression of PI3K signaling. Moreover, treatment of GNB2 KO cells with the p110alpha inhibitor BYL719 revealed increased dependence on p110alpha. To support clinical relevance, the authors also interrogated the METABRIC database and found GNB2 to be overexpressed in PTEN-low/mut breast cancers. A screen of GPCR inhibitors identified the PAR1 inhibitor vorapaxar as capable of phenocopying GNB2 KO. Signaling through PAR1 to AKT and ERK was negated in GNB2 KO cells, demonstrating the necessity of GNB2 in this context. The generalizability of this new treatment combination of vorapaxar and pan-PI3K or HER inhibitor was confirmed in a panel of PTEN-null TNBC cell lines.

The present manuscript does add to the body of literature further evidence of the utility of combination therapy to the PI3K and EGFR pathways in TNBC, and underscores the importance also of the G protein beta subunit and the potential for targeting G proteins in this aggressive subtype of breast cancer.

Minor concerns:

1. It would be interesting to test in vivo whether PAR1 inhibition is synergistic with HER or PI3K inhibition as shown in vitro.
2. In light of Filardo et al, Mol Endocrinol 2000 (PMID 11043579), have the authors considered GPER1 as an upstream effector in PTEN-null TNBC?
3. What is the target in Figure 4B to the far left?
4. Some relevant papers should be cited at relevant sections:
 - a. GCPR involvement in p110beta signaling: A novel role for phosphatidyl 3-kinase β in signaling from G protein-coupled receptors to Akt. Murga et al, J Biol Chem 2000 (PMID 10766839)
 - b. Combination therapy to EGFR and PI3K pathways in TNBC: Integrated molecular pathway analysis informs a synergistic combination therapy targeting PTEN/PI3K and EGFR pathways for basal-like breast cancer. She et al, BMC Cancer 2016 (PMID 27484095)
5. Figure 3 panels C and D have spliced images. Perhaps this should be made more evident, or a note added to the legend indicating that they are spliced from the same blots (my assumption).
6. Figure 1i and Figure 5e - please indicate the sample sizes.
7. In the authors' ESMO presentation <http://dx.doi.org/10.1136/esmoopen-2018-EACR25.20>, the CRISPR-Cas9 screen is described as having 144 genes, whereas this manuscript says 141 genes. Why the discrepancy?

Referee #3 (Comments on Novelty/Model System for Author):

Appropriate cell culture, xenograft, and GEMM models used, although I recommend one additional animal experiment.

Referee #3 (Remarks for Author):

Triple negative breast cancers are a significant minority of breast cancers that are difficult to control and often have active PI3K signaling. Pan-PI3K and isoform-selective PI3K inhibitors are under clinical investigation, but utility has been limited somewhat by toxicities. PTEN loss occurs in one-third of TNBC. The authors used consecutive knockdown and knockout screens to identify targets for combined therapy with PI3K pathway inhibitors in PTEN- TNBC. A genome-wide shRNA screen revealed that inhibition of EGFR cooperates with AZD8186(PI3Kbeta/delta), GDC0941pan-PI3Ki, and MK2206 (AKTi) in growth inhibition in tissue culture and an orthotopic xenograft. AZD8186 cooperates with gefitinib in reducing tumor growth in a WAP-CRE p53/PTEN mouse model. In tissue culture, p110beta, but not p110 alpha, association with EGFR is reduced. Decreased S6 phosphorylation was a common correlative endpoint for response in PTEN mutant/null in vitro and in vivo. CSNK2B and CSNK2A3 casein kinase genes were also hits in this screen and were validated.

A second DOX-inducible Cas9/ CRISPR screen used 141 candidates from the shRNA screen and proteins upregulated/downregulated by AZD4668 in RPPA assays. Hits included GNB2 and GNB5,

encoding beta and gamma G protein subunits. From a library of GPCR antagonists, Vorapaxar, which inhibits Thrombin receptor PAR1, suppressed P-Akt and P-S6 in combination with GDC or lapatinib. PAR1 induced p-Akt and p-S6, which was blocked by GNB2 knockout. In summary, the manuscript identified three different sets of therapeutic targets (EGFR, CSK2, Gbeta/gamma) that cooperate with PI3K pathway antagonists, and elucidated signaling mechanisms including activation of PI3K beta by EGFR (and reported earlier for Gbeta/gamma) and blockade of compensatory signaling through PI3Kalpha with GNB2 knockout, all in TNBC PTEN- backgrounds. This work is a significant advance in uncovering partner therapeutic targets for PI3K pathway inhibitors. The focus on PTEN-negative TNBC is important, as it identifies the core patient group for eventual clinical trials of these combinations. While EGFR amplification is especially prevalent in TNBC, EGFR activation by mutation or exposure to EGFR ligands or activation of other ERBBs, so these findings may extend to other PTEN-null solid tumors. Drugs targeting EGFR and PAR-1 that were used in this study are already in clinical use, so the translational path may be relatively short. The experimental work is technically excellent, although some general questions remain.

1. The most unexpected result is the impact of dual targeting with PAR1 inhibitor Vorapaxar. It's important to test the combination in an in vivo model, such as was done for EGFR combinations, given the usual issue that dual targeting increases likelihood of toxicities.
2. Xenograft and GEMM tumor studies in Figure 2 are carried out for four weeks or less. Do these tumor models eventually escape? (There is a three point upward trend in Fig 2E).
3. Two different supplementary table 1s (shRNA screen) are provided, one with GDC only, the other with the three agents. Which will be included with the final paper?
4. Relationship of PAR1 to totality of effects going through Gbeta/gamma would be clearer with PAR1 knockout experiments.

Minor and presentation issues.

1. How well do the new combinations work on TNBC with Plk3CA activating mutations?
2. Cantley has highlighted in vivo feedback activation of insulin receptor as a significant obstacle to effect of PI3K targeting (PMID 30158705). Comment on this in Discussion?
3. Several figures use micromolar and higher inhibitor concentrations, even for erlotinib. Is off-target inhibition a concern?
4. shRNA dose response curves: similar coloring and small symbols of different shRNAs makes it difficult to distinguish them.
5. Supp 2D: duration of experiment?
6. Fig. 6D: which cell lines used?
7. Typo: both expressing high levels of EGFR and showing different degree of sensitivity to PI3K inhibition in vitro (Suppl. Figure 1A). should be S2A

EMM-2020-11987 Zecchin et al.

Response to Reviewer #1

Response to Reviewers' Comments**EMBO Molecular Medicine EMM-2020-11987 Zecchin et al.**

21 April 2020

(Reviewers' original comments in italic, authors' responses in non-italic.)

Reviewer #1: Response to reviewers' comments

The manuscript by Zecchin et al describes the discovery of a number of potential additional targets that would be useful in PI3K inhibition in PTEN deficient triple negative breast cancer. Intriguingly, the authors find key role of GPCRs and EGFR, which in the context of the PTEN deficiency suggested a targetable strategy of inhibiting the beta isoform of PI3K beta in combination with these identified targets. This research provides an intriguing mechanism to develop novel therapeutic targeted combination strategies towards PTEN deficient TNBC. Overall, this research appears to be well described and would provide a useful advance to researchers both in the PI3K and TNBC fields.

This research is another excellent example of the ability of PI3K beta to synergize inputs from both RTK and GPCR inputs, and provides an exciting new rationale for designing novel combinational therapies.

My only true concern is in the true isoform selectivity of the AZD8186 compound, which should be able to be addressed in a very simple set of experiments.

We thank the referee for the appreciative words on our study. The isoform selectivity issue is addressed in response to point 1 below.

Major points

1. In the results text the AZD8186 inhibitor is described as beta isoform specific. The IC50 values of this compound have been reported (see below), and calling it beta specific might be a stretch. It inhibits delta with very similar IC50 values (although this isoform would likely not be expressed in these cells), with it being roughly 10 fold selective over alpha. This could likely be reworded to more clearly describe its specificity, as readers of this manuscript may not catch this subtlety.

We thank the reviewer for pointing out this issue and acknowledge that AZD8186 lacks strong specificity for beta over delta isoform of p110, as the IC50 of the compound is reported as 4nM for p110beta and 12nM for p110delta. We have changed the text to include a short description of the properties of AZD8186 in the first section of the results.

We also analysed RNAseq data from the Broad Institute CCLE (Cancer Cell Line Encyclopedia) database (<https://portals.broadinstitute.org/ccle>), comparing expression of the genes encoding the four p110 isoforms (PIK3CA, PIK3CB, PIK3CD and PIK3CG) between the six PTEN-null breast cancer cell lines used in this study and eleven lines from B-cell acute lymphoblastic leukemia (B-ALL), which are known to express all four p110 isoforms, including p110delta and gamma (Thorpe LM et al, 2015). As expected, this comparison shows that p110delta, and also p110gamma, is not expressed or expressed at very low levels in PTEN-null breast cell lines, in contrast to the alpha and beta isoforms (EV Fig 1B). Therefore, we concluded that it is very unlikely that the effects observed in vitro following AZD8186 treatment may be due to p110delta inhibition.

The authors do note that they use a concentration in cells that will mainly target p110 beta derived from Schwartz et al.

<https://mct.aacrjournals.org/content/14/1/48.long>

<https://www.sciencedirect.com/science/article/pii/S1535610814004590?via%3Dihub>

In this case the authors of that 2015 manuscript determined that AZD8186 had no effect at up to

250 nM in a HER2 p110alpha dependent tumor, so at this concentration it is mainly exerting its effect on p110 beta. This strikes me that this will be cell type dependent based on the actual amount of p110 alpha vs p110 beta that is expressed in the given cell type. These cells likely will not have delta expression, so the determination of the role of p110 alpha vs beta is critical. It would be useful in one of these cell models (the simplest might be to repeat the work in Fig 1G) to use in addition a p110 alpha specific inhibitor in combination with an EGFR inhibitor to verify the importance of p110 beta. This does not need to be repeated across all cell types, but would reinforce the cited work by Schwartz et al, and support the claims of the important role of p110 beta. The authors have already used this compound in the Fig 5 in the GNB2 knockout.

Following the reviewer's suggestion, we have tested BYL719, a p110alpha specific inhibitor, alone or in combination with the EGFR inhibitor gefitinib on three PTEN-null and three PTEN-WT triple-negative breast cancer cell lines among the ones employed in this study. In parallel, we also used AZD8186, alone or in combination with gefitinib, again on the same cell lines, as a control for p110beta inhibition. The results are reported in EV Fig 1G: they demonstrate that BYL719 (1.2µM), even when combined with gefitinib, does not produce any selective anti-proliferative effect on PTEN-null cells compared to PTEN WT lines, while the combination of AZD8186 (90 nM) with gefitinib does, confirming our previous observations. On the contrary, p110alpha inhibition produced a pronounced anti-proliferative effect when combined with EGFR inhibitor on one of the two PIK3CA-mutant triple negative breast cell lines tested in parallel.

These results confirmed that inhibition of p110alpha is not responsible for the genotype-selective and synergistic effects of AZD8186 and gefitinib combined treatment in PTEN-null triple negative breast cancer cells.

However, in the mouse experiments (Fig. 2) it is critical to note that the AZD8186 compound will target delta almost as effectively as beta (4 vs 12 nM). While I do not think it is necessary to repeat any of the mouse studies with a delta inhibitor, it is essential that they at least clearly describe this potential complication of the experiment in the results and discussion, as delta inhibition can have other anti-tumor effects.

We appreciate referee's concern and agree that it is difficult to formally exclude at this stage that AZD8186 activity in vivo may be partially due to targeting of p110delta-expressing cells— especially immune-cells characterised by high levels of p110delta expression. However, we confirmed anti-tumour activity and lack of toxicity of the combination AZD8186 – erlotinib in both immune-competent and immune-deficient mice, and we believe that this evidence makes less likely that AZD8186-mediated manipulation of the immune-compartment had a key role in determining the response. This issue has now been discussed in the results and discussion sessions of the manuscript, as suggested by the reviewer.

2. The discovery of the CK2 kinase role in this process is intriguing and unexpected. Can the authors expand in the discussion on the possible role of this kinase in the pathway in the discussion. Currently this is brought up in the results, and not explored in depth in the discussion.

We have now included in the manuscript a paragraph discussing the potential role of CK2 in PI3K-AKT pathway activation (Discussion section).

3. There are a number of comments in the discussion that state 'for the first time ...'. I am not sure these novelty phrases improve the work, and I think they should be removed. We have removed those novelty phrases from the discussion.

Reviewer #2: Response to reviewers' comments

This well-written and structured manuscript represents an impressive series of experiments which highlight the importance of two signaling branches, EGFR and GPCR, in PTEN-deficient TNBC using relevant models, and which may have clinical relevance in the future. Zecchin et al have performed a whole genome shRNA screen in a PTEN-null TNBC model to identify targets whose inhibition synergizes with inhibitors of PI3K pathway components, namely AZD8186 (inhibiting p110beta), GDC0941 (pan-PI3K), and MK2206 (AKT), followed by a comprehensive series of mechanistic studies in relevant models and validation work. This well-written and structured manuscript represents an impressive series of experiments which highlight the importance of two signaling branches, EGFR and GPCR, in PTEN-deficient TNBC using relevant models, and which may have clinical relevance in the future. Among the shRNA screen hits, the authors focused on EGFR, and the casein kinase 2 components CSNK2B and CSNK2A2, which were selectively depleted in the AZD8186 condition compared to untreated control. Combination treatment of cells with EGFR-inhibitors and PI3K pathway inhibitors was synergistic, as was the combination of EGFR-inhibitors and PI3Kbeta inhibition in human xenografts and in an immunocompetent conditional Pten/p53-null mouse model and isografts. To support clinical relevance, the authors also interrogated the METABRIC database and found EGFR to be overexpressed in PTEN-low/mut breast cancers. To further dissect the signaling downstream of combination treatments, western blotting experiments across PTEN-null cell lines and treatment conditions identified decreased phospho-S6 as a good biochemical readout of response. Co-IP experiments under various treatments showed p110beta to interact with EGFR, which was disrupted most potently by combined p110beta and EGFR inhibition, and that AKT phosphorylation downstream of EGFR signaling was mediated in part by p110beta. Furthermore, to validate the screen and identify additional modifiers, 110 of the shRNA hits plus 31 hits from RPPA analysis of MDA-468 cells treated with AZD8186, were used in a CRISPR-Cas9 screen using pS6 as the readout. The shRNA screen was largely validated; and the authors identified knock out of GNB2 and GNG5 (encoding G protein beta and gamma subunits) as cooperative partners of GDC0941 pan-PI3K inhibition. KO clones of GNB2 had increased EGFR expression, phosphorylation, and proportion of phospho-EGFR, and decreased pAKT and pS6 which was accentuated with GDC0941 treatment. EGFR-inhibition in the GNB2 KO context also led to greater suppression of PI3K signaling. Moreover, treatment of GNB2 KO cells with the p110alpha inhibitor BYL719 revealed increased dependence on p110alpha. To support clinical relevance, the authors also interrogated the METABRIC database and found GNB2 to be overexpressed in PTEN-low/mut breast cancers. A screen of GPCR inhibitors identified the PAR1 inhibitor vorapaxar as capable of phenocopying GNB2 KO. Signaling through PAR1 to AKT and ERK was negated in GNB2 KO cells, demonstrating the necessity of GNB2 in this context. The generalizability of this new treatment combination of vorapaxar and pan-PI3K or HER inhibitor was confirmed in a panel of PTEN-null TNBC cell lines. The present manuscript does add to the body of literature further evidence of the utility of combination therapy to the PI3K and EGFR pathways in TNBC, and underscores the importance also of the G protein beta subunit and the potential for targeting G proteins in this aggressive subtype of breast cancer.

We are grateful to the referee for his/her positive comments on our work.

Minor concerns:

1. It would be interesting to test in vivo whether PAR1 inhibition is synergistic with HER or PI3K inhibition as shown in vitro.

In response to this suggestion, we performed the experiment by testing GDC0941 pan-PI3K inhibitor (100mg/kg) and the PAR1 inhibitor Vorapaxar (30mg/kg), alone or in combination, on MDA-MB-468 mammary fat-pad tumour xenografts for 17 days. The results are reported in figures below for the reviewer below. They show a trend toward a stronger anti-tumour effect for the combination compared to single-agent treatments (Referee #2 Figure 1), with no toxic effects observed, as shown by unchanged mouse weight during the treatment (Referee #2

Figure 2). The difference between treatments, however, is not statistically significant and the effect is not as clear as the one observed for combined inhibition of EGFR and PI3K β .

Due to the tendency of MDA-MB-468 tumours treated by GDC0941 to lose their solid consistency and “liquefy”, we were prompted in this specific case to measure the response to the drugs with a method that was not dependent on the consistency of the tumour masses. Indeed, in order to eliminate any possible bias between different treatment groups, we randomised mice based on tumour volumes before starting the treatments and, to evaluate response to the drugs, we weighed explanted tumours at the end of the treatments. Despite the lack of high sensitivity for this type of measurements, we could still observe a trend toward a more prominent response in the combination group, although we acknowledge that differences may have been partially masked.

Although we cannot formally rule out that insufficient target inhibition by vorapaxar within the tumour mass may have occurred during the treatment, we tried to circumvent this problem by using in this experiment a high dose of vorapaxar, being 25mg/kg/day the reported NOAEL (no-observed-adverse-effect-level) over three months treatment in mice (FDA report at https://www.accessdata.fda.gov/drugsatfda_docs/nda/2014/204886Orig1s000PharmR.pdf). Indeed, 15mg/kg/day, based on plasma exposure to vorapaxar for mice, is equivalent to 30 times the recommended therapeutic exposures in humans and inhibition of PAR-1-dependent phenotypes in mice with doses of vorapaxar as low as 5 μ g and 10 μ g/kg have been previously reported (Noguchi D et al. 2020, Kim, H. N., et al. 2015).

However, we have previously discussed in the manuscript that vorapaxar treatment in vitro by its own, even when used at concentration that proved to completely prevent the PAR1 agonist-mediated activation of AKT and ERK (Fig. 6c), was not sufficient to partially decrease the phosphorylation of AKT or S6 (Fig. 6b), which is different to knock out of GNB2 (Fig. 4d). This suggests that other GPCRs or compensation mechanisms may act upstream of GNB2 and PI3K β , and, therefore, it is conceivable that the combinatorial inhibition of PAR1 and PI3K or EGFR may not be as efficient and selective as drug combinations including PI3K β inhibitor.

Altogether our data show a mechanistic role of PAR1 upstream of G β γ subunit in supporting activation of PI3K β and in preventing response to targeting of the pathway by PI3K or EGFR inhibitors. However, our current data do not suggest that therapeutic use of combinatorial therapies including the PAR1 inhibitor vorapaxar may produce similar or stronger response compared to EGFR-PI3K β combined inhibition. As the in vivo effects on breast tumour growth of vorapaxar, either alone or in combination with GDC0941, are not statistically significant, we do not plan to include these data in the manuscript, but we do discuss this further in the Discussion section.

Referee #2 Figure 1. Tumor weight of MDA-MB-468 mammary fat-pad xenografts treated with vehicle, Vorapaxar (30 mg/kg, og once/day), GDC0941 (100 mg/kg IP once/day) alone or in combination (6-7 mice per group, median, interquartile change) for 17 days. Differences were non statistically significant (unpaired t test).

Referee #2 Figure 2. Change in the body weight of mice harbouring MDA-MB-468 xenograft tumors during 12 days of treatment.

2. *In light of Filardo et al, Mol Endocrinol 2000 (PMID 11043579), have the authors considered GPER1 as an upstream effector in PTEN-null TNBC?*

We thank the referee for this suggestion. We checked data from our high throughput screenings looking for information concerning GPER1. Unfortunately, the gene wasn't included in the list of about 16000 genes targeted by the shRNA library that was used in the initial genome-wide screening, and we cannot draw conclusions about its involvement in the signalling upstream PI3K β from our data. However, we acknowledged the potential involvement of GPER1 or other GPCRs apart from PAR1 in signalling upstream G $\beta\gamma$, -PI3K β in the discussion of the manuscript and we discussed how this may explain the difference in inhibition of PI3K downstream effectors (e.g. phospho-S6) observed following the sole inhibition of PAR1 or GNB2 KO (see also previous point).

3. *What is the target in Figure 4B to the far left?*

The target on the far left of the dot plot in Figure 4B is NDNL2. Although the amplitude of the effect was big, the difference between the effect observed in absence or presence of GDC0941 was not statistically significant ($p=0.07$). For this reason, we did not highlight the dot on the graph and we decided to do not further follow up this target.

4. *Some relevant papers should be cited at relevant sections:*

- a. *GCPR involvement in p110beta signaling: A novel role for phosphatidyl 3-kinase β in signaling from G protein-coupled receptors to Akt. Murga et al, J Biol Chem 2000 (PMID 10766839).*
- b. *Combination therapy to EGFR and PI3K pathways in TNBC: Integrated molecular pathway analysis informs a synergistic combination therapy targeting PTEN/PI3K and EGFR pathways for basal-like breast cancer. She et al, BMC Cancer 2016 (PMID 27484095)*

We thank the referee for this comment. We have added the suggested references to relevant sections in the revised discussion.

5. *Figure 3 panels C and D have spliced images. Perhaps this should be made more evident, or a note added to the legend indicating that they are spliced from the same blots (my assumption).*

We apologise for this oversight. The blots mentioned were cropped from the same images. We have now outlined the spliced images with black borders and have described this in the figure legend.

6. *Figure 1i and Figure 5e - please indicate the sample sizes.*

We have now added the sample sizes in the relevant figures mentioned.

7. *In the authors' ESMO presentation <http://dx.doi.org/10.1136/esmoopen-2018-EACR25.20>, the CRISPR-Cas9 screen is described as having 144 genes, whereas this manuscript says 141 genes. Why the discrepancy?*

We thank the referee for noticing the inconsistency. We confirm that the number of genes screened is 141. In the EACR 2018 presentation we counted also controls of the experiment, including two different non-target sgRNA groups and cells not-transduced with sgRNAs. We apologise for any confusion this may have generated.

Reviewer #3: Response to reviewers' comments

Triple negative breast cancers are a significant minority of breast cancers that are difficult to control and often have active PI3K signaling. Pan-PI3K and isoform-selective PI3K inhibitors are under clinical investigation, but utility has been limited somewhat by toxicities. PTEN loss occurs in one-third of TNBC. The authors used consecutive knockdown and knockout screens to identify targets for combined therapy with PI3K pathway inhibitors in PTEN- TNBC. A genome-wide shRNA screen revealed that inhibition of EGFR cooperates with AZD8186(PI3Kbeta/delta), GDC0941pan-PI3Ki, and MK2206 (AKTi) in growth inhibition in tissue culture and an orthotopic xenograft. AZD8186 cooperates with gefitinib in reducing tumor growth in a WAP-CRE p53/PTEN mouse model. In tissue culture, p110beta, but not p110 alpha, association with EGFR is reduced. Decreased S6 phosphorylation was a common correlative endpoint for response in PTEN mutant/null in vitro and in vivo. CSNK2B and CSNK2A3 casein kinase genes were also hits in this screen and were validates.

A second DOX-inducible Cas9/ CRISPR screen used 141 candidates from the shRNA screen and proteins upregulated/downregulated by AZD4668 in RPPA assays. Hits included GNB2 and GNB5, encoding beta and gamma G protein subunits. From a library of GPCR antagonists, Vorapaxar, which inhibits Thrombin receptor PAR1, suppressed P-Akt and P-S6 in combination with GDC or lapatinib. PAR1 induced p-Akt and p-S6, which was blocked by GNB2 knockout. In summary, the manuscript identified three different sets of therapeutic targets (EGFR, CSK2, Gbeta/gamma) that cooperate with PI3K pathway antagonists, and elucidated signaling mechanisms including activation of PI3K beta by EGFR (and reported earlier for Gbeta/gamma) and blockade of compensatory signaling through PI3Kalpha with GNB2 knockout, all in TNBC PTEN- backgrounds.

This work is a significant advance in uncovering partner therapeutic targets for PI3K pathway inhibitors. The focus on PTEN-negative TNBC is important, as it identifies the core patient group for eventual clinical trials of these combinations. While EGFR amplification is especially prevalent in TNBC, EGFR activation by mutation or exposure to EGFR ligands or activation of other ERBBs, so these findings may extend to other PTEN-null solid tumors. Drugs targeting EGFR and PAR-1 that were used in this study are already in clinical use , so the translational path may be relatively short.

The experimental work is technically excellent, although some general questions remain. Appropriate cell culture, xenograft, and GEMM models used, although I recommend one additional animal experiment.

1. The most unexpected result is the impact of dual targeting with PAR1 inhibitor Vorapaxar. It's important to test the combination in an in vivo model, such as was done for EGFR combinations, given the usual issue that dual targeting increases likelihood of toxicities.

We thank the referee for pointing out the relevance of our approaches and we performed an additional in vivo experiment as requested. In order to answer this question, we treated immune-compromised mice harbouring mammary fat pad MDA-MB-468 tumour xenografts with vehicle or GDC0941 (100 mg/kg/day, oral gavage) or vorapaxar (30 mg/kg/day, oral gavage) or a combination of the two drugs for 17 days. The results are shown in Referee #3 Figures 1 and 2 at the end of this paragraph. It is worth to notice that no significant changes in body weights (Referee #3 Figure 1) and no other toxic effects were observed on treating these mice, suggesting that vorapaxar alone or in combination with GDC0941 was well tolerated in vivo, even when used at relatively high doses in this experiment (see below).

Mice were randomised just before starting the treatment based on their tumour volumes and the xenografts were explanted and weighted at the end of the experiment to estimate the anti-tumour effects of the treatments. We chose this method to measure tumour response instead of evaluating changes in tumour volumes as treatment by GDC0941, alone or in combination, induced a loss of solid

consistency and “liquefied” MDA-MB-468 tumours. We decided, then, to apply a method that could estimate response to the drugs independently from the consistency of the tumour masses. We observed a trend towards a stronger response in the combination group compared to other treatment conditions, but the difference was not statistically significant (Referee #3 Figure 2). As these experiments did not give significant differences, we were not planning to present the data in the paper, although they are mentioned as negative results in the discussion.

Given the paucity of previous reports using vorapaxar in mouse models, we used a high dose, since 25mg/kg/day was reported as NOAEL (no-observed-adverse-effect-level) over three months treatment in mice (FDA report at https://www.accessdata.fda.gov/drugsatfda_docs/nda/2014/204886Orig1s000PharmR.pdf). Also, 15mg/kg/day, based on plasma exposure to vorapaxar for mice, is equivalent to 30 times the recommended therapeutic exposures in humans and inhibition of PAR-1-dependent phenotypes in mice with doses of vorapaxar as low as 5µg and 10µg/kg/day has been previously reported (Noguchi D et al. 2020, Kim, H. N., et al. 2015). This evidence makes it unlikely that vorapaxar used at 30mg/kg/day failed to inhibit its specific target within the tumours.

It is conceivable that PAR1 inhibition may not be as potent as inhibition of $\beta\gamma$ subunits of G protein or PI3K β downstream, and therefore, that combinatorial treatments including PAR1 inhibitors may not be as effective and specific as combinations with anti-PI3K β drugs. This hypothesis is supported by the evidence that vorapaxar treatment alone in vitro, even when used at concentrations that were able to completely prevent the PAR1 agonist-mediated activation of AKT and ERK (Fig. 6c), was not sufficient to partially decrease the phosphorylation of AKT or S6 (Fig. 6b), which was in contrast to GNB2 knock out (Fig. 4d). This may be due to redundancy of PAR1 and other GPCRs functions upstream of the G $\beta\gamma$ subunits, such that targeting no single GPCR can have a strong an effect as targeting the G $\beta\gamma$ downstream mediators. This issue has now been discussed in the revised version of the manuscript.

Overall, our data, although they support a mechanistic role of PAR1 upstream of the $\beta\gamma$ subunits of G proteins and PI3K β in preventing response to PI3K pathway inhibitors, do not recommend combinatorial treatments with vorapaxar as an alternative therapeutic option to PI3K β and EGFR combined targeting. We envision that this evidence will foster further investigation of the GPCRs signaling involved in the resistant phenotypes here described and, given the lack of toxicity so far observed, this will encourage the identification of suitable anti-GPCR drugs with strong therapeutic impact.

Referee #3 Figure 1. Change in the body weight of mice harbouring MDA-MB-468 xenograft tumors during 12 days of treatment.

Referee #3 Figure 2. Tumor weight of MDA-MB-468 mammary fat-pad xenografts treated with vehicle, Vorapaxar (30 mg/kg, og once/day), GDC0941 (100 mg/kg IP once/day) alone or in combination (6-7 mice per group, median, interquartile change) for 17 days. Differences were non statistically significant (unpaired t test).

2. *Xenograft and GEMM tumor studies in Figure 2 are carried out for four weeks or less. Do these tumor models eventually escape? (There is a three point upward trend in Fig 2E).*

We did not test treatments for periods longer than the ones reported in the manuscript. However, the waterfall plot reported in EV Fig 2A showed that just a fraction of individual tumours (2/6) responded differently and less dramatically to the combinatorial treatment compared to the others. Therefore, we predict that over a long period of treatment it is likely that a portion of tumours could relapse, similarly to what has been observed for most of the targeted therapies producing anti-tumour effects tested so far.

3. *Two different supplementary table 1s (shRNA screen) are provided, one with GDC only, the other with the three agents. Which will be included with the final paper?*

We thank the reviewer for this observation and we apologize for the oversight. The error occurred during the conversion of the file and in the revised version of the manuscript now includes genes ranks for the three drugs.

4. *Relationship of PAR1 to totality of effects going through Gbeta/gamma would be clearer with PAR1 knockout experiments.*

We agree that this would add further useful evidence to our paper. Unfortunately, although we started the procedure to derive PAR1 KO clones from MDA-MB-468, we realised that generation of validated KO cells may prove longer than expected. This is due to technical difficulties in screening and functional validation of PAR1 KO clones, related to the lack of reliable PAR1 antibodies and not fully elucidated functional properties of PAR1 in cancer cells, and also to the peculiar circumstances that are currently preventing the execution of further lab work.

However, we hope that the variety of experimental approaches here presented to answer the scientific question from different angles, including unbiased drug screening (Fig.6A), different combinatorial drug treatments in multiple cell models (Fig. 6B-D) and comparison of agonist-mediated activation of PAR1 in WT and GNB2 KO cells (Fig.6C) provides sufficient evidence of involvement of PAR1 in the signalling network described. A quantification of the increase in p-AKT and p-ERK in MDA-MB-468 WT and GNB2 KO for the blot in Fig. 6C has been also added to the revised manuscript for clarity.

Minor and presentation issues.

1. *How well do the new combinations work on TNBC with PIK3CA activating mutations?*

We thank the referee for opening a new interesting angle for our research. PIK3CA mutations are reported in around 10-18% of TNBCs in TCGA and in a Chinese TNBC cohort, respectively (Jiang, Y. Z. et al. 2019). We identified only two PIK3CA-mutant cell lines, namely BT20 and SUM159PT, and we tested anti-proliferative effects of treatment by PI3K β inhibitor AZD8186 or PI3K α inhibitor BYL719 alone or in combination with EGFR inhibitor gefitinib. Results were included in the EV Fig 1G and show no effect for AZD8186, alone or added to EGFR inhibitor, in PIK3CA-mutant TNBC cells. However, one of the two cell lines tested, BT20, was partially responsive to BYL719 and gefitinib single-agent treatments and more responsive to the two drugs combined. Although, given the paucity of TNBC cell models harbouring PIK3CA mutations, these results are not conclusive, they still suggest that a sub-population of those tumours may be responsive to the combinatorial targeting of PI3K α and EGFR.

2. *Cantley has highlighted in vivo feedback activation of insulin receptor as a significant obstacle to effect of PI3K targeting (PMID 30158705). Comment on this in Discussion?*

We thank the referee for commenting on this. We believe that specific targeting of β over α isoform of PI3K, in combination with EGFR inhibition, may not trigger the resistance mechanism described by Cantley and colleagues, that is mediated by feedback reactivation of insulin pathway following PI3K α blockade and inhibition of glucose uptake. This may not happen in the context of PI3K β inhibition, since hyperglycaemia, that in turn would trigger increased insulin secretion in the blood stream, was not reported among the adverse effects in patients treated by this drug (Lillian S. et al, 2016). We have discussed this interesting aspect in the revised version of the manuscript. However, we cannot exclude that physiological fluctuations of insulin levels may partially reactivate PI3K β in cancer cells through insulin receptor activation, overriding the effects of PI3K β inhibitor. Therefore, it would be worth to further investigate the benefit of a controlled diet in this context of PI3K β inhibitor treatment.

3. *Several figures use micromolar and higher inhibitor concentrations, even for erlotinib. Is off-target inhibition a concern?*

We appreciate the referee's concerns about the use in this study of high concentrations of some compounds, in particular erlotinib. Indeed, we acknowledge that erlotinib was reported to produce some off-target effects at those concentration (Karaman M.W. et al 2008, Yamamoto N. et al 2011, Conradt L et al. 2011). However, we believe that the use of multiple specific EGFR inhibitors such as gefitinib and cetuximab or pan-HER specific inhibitor lapatinib to confirm our findings rules out the possibility that the effects observed may be due to off-target inhibition.

4. *shRNA dose response curves: similar coloring and small symbols of different shRNAs makes it difficult to distinguish them.*

We apologise for any confusion that the color-coding and symbols may have created. Figures 1D and EV Fig. 1J have been changed to make them clearer.

5. *Supp 2D: duration of experiment?*

Apologies for this oversight. The treatment lasted 21 days and this information has been added in the legend of EV Fig 2D for clarity.

6. *Fig. 6D: which cell lines used?*

The cell lines used in the experiments reported in Fig 6D, and also those used in Fig 1G, have now been listed in the legends of the respective figures.

7. *Typo: both expressing high levels of EGFR and showing different degree of sensitivity to PI3K inhibition in vitro (Suppl. Figure 1A). should be S2A.*

We thank the referee for noticing this; we apologise for the confusion generated by our phrasing and we have edited the sentence to make it clear.

REFERENCE LIST

- Conradt, L., et al. (2011). "Disclosure of erlotinib as a multikinase inhibitor in pancreatic ductal adenocarcinoma." *Neoplasia* **13**(11): 1026-1034.
- Jiang, Y. Z., et al. (2019). "Genomic and Transcriptomic Landscape of Triple-Negative Breast Cancers: Subtypes and Treatment Strategies." *Cancer Cell* **35**(3): 428-440 e425.
- Karaman, M. W., et al. (2008). "A quantitative analysis of kinase inhibitor selectivity." *Nat Biotechnol* **26**(1): 127-132.
- Kim, H. N., et al. (2015). "Protease activated receptor-1 antagonist ameliorates the clinical symptoms of experimental autoimmune encephalomyelitis via inhibiting breakdown of blood-brain barrier." *J Neurochem* **135**(3): 577-588.
- Lillian, S. et al. (2016) "AZD8186 study 1: Phase I study to assess the safety, tolerability, pharmacokinetics (PK), pharmacodynamics (PD) and preliminary anti-tumour activity of AZD8186 in patients with advanced castration-resistant prostate cancer (CRPC), squamous non-small cell lung cancer, triple negative breast cancer and with PTEN-deficient/mutated or PIK3CB mutated/amplified malignancies, as monotherapy and in combination with vistusertib (AZD2014) or abiraterone acetate." *European Journal of Cancer*, Volume 69, S19
- Noguchi, D., et al. (2020). "Antiapoptotic Effect by PAR-1 Antagonist Protects Mouse Liver Against Ischemia-Reperfusion Injury." *J Surg Res* **246**: 568-583.
- Thorpe, L. M., et al. (2015). "PI3K in cancer: divergent roles of isoforms, modes of activation and therapeutic targeting." *Nat Rev Cancer* **15**(1): 7-24.
- Yamamoto, N., et al. (2011). "Off-target serine/threonine kinase 10 inhibition by erlotinib enhances lymphocytic activity leading to severe skin disorders." *Mol Pharmacol* **80**(3): 466-475.

16th May 2020

Dear Dr. Downward,

Thank you for the submission of your revised manuscript to EMBO Molecular Medicine. We have now received the enclosed reports from the three referees who reviewed the new version of your manuscript. As you will see, they are now supportive of publication, and I am thus pleased to inform you that we will be able to accept your manuscript pending the following final editorial amendments:

1) Main manuscript text:

- Please correct/answer the track changes suggested by our data editors in the main manuscript file (in track changes mode). I will send you the Word file in the next couple of days.
- Please provide up to 5 keywords.
- Please move the Material and Methods section up in the main file, so that it is placed after the discussion.
- Author contribution: please complete (Sareena Rana missing).
- Conflict of interest: this section appears twice in the manuscript (p. 23 and p. 34), please only keep a "conflict of interest" section p. 34.
- Data availability section: Thank you for depositing your data in a public repository. Please note that the data under the access code GSE148785 is currently not accessible (public release scheduled on Apr 15, 2023). This data should be made available to the public before acceptance of the manuscript.
- Please indicate in legends or in the figures the exact $n=$ and exact $p=$ values, not a range, along with the statistical test used. Some people found that to keep the figures clear, providing a supplemental table with all exact p -values was preferable. You are welcome to do this if you want to.
- In the Material and Methods section, please indicate the antibody dilutions.

2) Figures:

- Figure 3C: please check the labelling of the blots, one letter ("H" from "VEH") appears on the blot.
- Please indicate in the legends when blots have been cut and reassembled for figure purposes (such as Fig. 3D P6).
- Please add scale bars to figures EV2B, EV4K, EV5D.
- Please add space between the different pictures in Fig. 6E.
- Figure callouts: The main text refers to panels Fig. 1J-M, which do not exist, please correct.
- The legends for EV Tables should be added directly to the tables (top of the page or separate tab). The EV figure legends should be added to the main manuscript text, after the main figure legends.

3) Source data:

Thank you for providing Source Data for your figures. Please upload your source data so as to have one file per figure. Labelling the blots would help navigate the western blots source data.

4) Synopsis:

Thank you for providing a synopsis text. Please also suggest a striking image or visual abstract to illustrate your article as a jpeg or png file 550 px-wide x 400-px high.

5) As part of the EMBO Publications transparent editorial process initiative (see our Editorial at <http://embomolmed.embopress.org/content/2/9/329>), EMBO Molecular Medicine will publish online a Review Process File (RPF) to accompany accepted manuscripts.

In the event of acceptance, this file will be published in conjunction with your paper and will include the anonymous referee reports, your point-by-point response and all pertinent correspondence relating to the manuscript. Let us know whether you agree with the publication of the RPF and as here, IF YOU WANT TO REMOVE OR NOT any figures from it prior to publication.

I look forward to receiving your revised manuscript.

Yours sincerely,

Lise Roth

Lise Roth, PhD
Editor
EMBO Molecular Medicine

To submit your manuscript, please follow this link:

Link removed

The system will prompt you to fill in your funding and payment information. This will allow Wiley to send you a quote for the article processing charge (APC) in case of acceptance. This quote takes into account any reduction or fee waivers that you may be eligible for. Authors do not need to pay any fees before their manuscript is accepted and transferred to our publisher.

***** Reviewer's comments *****

Referee #1 (Remarks for Author):

The authors have done an excellent job addressing my concerns, and this paper will make a significant contribution to our understanding of targeting PI3K in disease.

Referee #2 (Remarks for Author):

Thank you for the comprehensive responses to my and the other reviewers' comments. To this

reviewer, the concerns have been adequately addressed.

Referee #3 (Comments on Novelty/Model System for Author):

appropriate cell culture, xenograft, and genetically-engineered models used

Referee #3 (Remarks for Author):

The authors have responded constructively to issues identified in the original manuscript, including substantial experimental work.

Corresponding Author Name: Julian Downward

Manuscript Number: EMM-2020-11987